# R1-Code-Interpreter: LLMs Reason with Code via Supervised and Multi-stage Reinforcement Learning

**Yongchao Chen**
MIT / Harvard
yongchaochen@fas.harvard.edu

**Yueying Liu**
University of Illinois Urbana-Champaign
yl136@illinois.edu

**Junwei Zhou**
University of Michigan
zhoujw@umich.edu

**Yilun Hao**
MIT
yilunhao@mit.edu

**Jingquan Wang**
University of Wisconsin–Madison
jwang2373@wisc.edu

**Yang Zhang**
MIT-IBM Watson AI Lab
Yang.Zhang2@ibm.com

**Na Li**
Harvard
nali@seas.harvard.edu

**Chuchu Fan**
MIT
chuchu@mit.edu

## Abstract

Practical guidance on training Large Language Models (LLMs) to leverage Code Interpreter across diverse tasks remains lacking. We present R1-Code-Interpreter, an extension of a text-only LLM trained via multi-turn supervised fine-tuning (SFT) and reinforcement learning (RL) to autonomously generate multiple code queries during step-by-step reasoning. Unlike prior RL + tool-use efforts focused on narrow domains such as math or retrieval, we curate 144 diverse reasoning and planning tasks and show that training a general-purpose Code Interpreter across them presents significant challenges due to task heterogeneity and scarcity of effective samples. To address this, we introduce a multi-stage curriculum learning approach that partitions training samples by measured improvement potential. The RL training prioritizes samples with higher potential and gradually shifts to lower-potential ones, increasing the average RL gains from merely +3.4% to +9.3% across Qwen-2.5 models (3/7/14B). Our final model, R1-CI-14B, improves average accuracy on the 37 test tasks from 44.1% to 72.4%, outperforming text-only GPT-4o (58.6%) and GPT-4o with Code Interpreter (70.9%). Notably, R1-CI-14B also exhibits emergent self-checking behavior through code generation. Datasets, Codes, and Models are available at https://github.com/yongchao98/R1-Code-Interpreter and https://huggingface.co/yongchao98.

## 1 Introduction

While reinforcement learning (RL)-based fine-tuning has significantly improved LLMs' reasoning and planning Wang et al. (2024); Guo et al. (2025); Jaech et al. (2024), models still struggle with seemingly simple tasks (Chen et al., 2025) and incur high token costs during inference-time search (Chen et al., 2024a). Textual reasoning excels at semantics and commonsense, but falls short in precise computation, symbolic manipulation, optimization, and algorithmic processing (Valmeekam et al., 2022). In contrast, symbolic code generation handles these rigorously and benefits from external tools (e.g., equation solvers). Prompting LLMs to generate and execute code often outperforms pure textual reasoning (Madaan et al., 2022; Liang et al., 2022; Chen et al., 2022).

A key challenge is guiding LLMs to decide when to rely on textual reasoning versus programmatic solutions, given that most input questions lack explicit cues about which approach is best and the possible text/code solution space is large. OpenAI's GPT models address this by incorporating a Code Interpreter, allowing iterative code generation and reasoning over outputs (Achiam et al.,

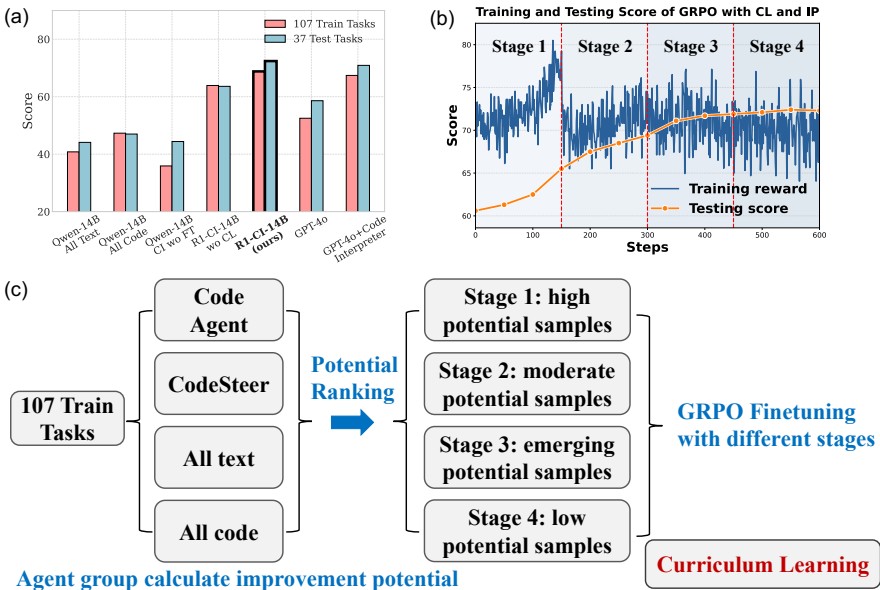

Figure 1: Training Code Interpreter-augmented reasoning models with multi-stage GRPO on 144 reasoning and planning tasks. (a) Our best model, R1-CI-14B, outperforms both GPT-4o (text-only) and GPT-4o with Code Interpreter. (b) Training reward and test scores improve steadily through the curriculum learning, then plateau at stage 4 after adding low-potential samples. (c) To assess sample effectiveness, we estimate improvement potential by repeatedly sampling answers with different agent frameworks and analyzing the correct/wrong distribution. GRPO begins with high-potential samples and gradually incorporates lower-potential ones.

2023). However, recent work Chen et al. (2024c) show that current Code Interpreter implementations struggle to effectively steer between text and code, underutilizing symbolic capabilities. Recent work such as ToRL (Li et al., 2025b) and ReTool (Feng et al., 2025) investigates training reasoning models to integrate with Code Interpreters. However, their training and evaluation are limited to math problems, leaving a significant gap from real-world applications that demand effectiveness across broader benchmarks. ToolRL (Qian et al., 2025) instead focuses on teaching models to select among multiple tools, where the Code Interpreter is used only for generating relatively simple code. Currently, public research still lacks a comprehensive understanding of how to fine-tune LLMs to integrate with Code Interpreter for robust, generalizable performance across **hundreds of tasks**.

To tackle these challenges, we present R1-Code-Interpreter, a framework for integrating Code Interpreter into open-source LLMs. We curate 144 reasoning and planning tasks and synthesize 6.5k multi-turn text/code trajectories for supervised fine-tuning (SFT), followed by Group Relative Policy Optimization (GRPO) (Shao et al., 2024). In contrast to prior work, which shows that RL training can yield substantial improvements on single or simple tasks, we find that **applying traditional DeepSeek-style RL training to a general Code Interpreter across 144 challenging tasks yields only marginal gains**. This difficulty arises from task heterogeneity and the scarcity of effective samples. To address this, we propose a novel multi-stage curriculum learning approach guided by measured *improvement potential*. Specifically, after the SFT stage, the model is prompted with diverse single- and multi-turn agent strategies to answer the same question. The accuracy discrepancies among these answers are used to estimate improvement potential. RL training then proceeds in four stages, beginning with high-potential samples and moving to lower-potential ones.

We finetune Qwen-2.5 (Qwen et al., 2025) models (3/7/14B), achieving average success rate improvements of 33.7% on 107 train tasks and 34.1% on 37 test tasks. As shown in Fig. 1, our best model, R1-CI-14B, raises testing accuracy from 44.1% to 72.4%, outperforming much larger GPT-4o text-only (58.6%) and GPT-4o with its inherent Code Interpreter (70.9%). The resulting model effectively combines code execution with textual reasoning, solving tasks through iterative "execute-and-explore"

interactions before producing final answers. The model gradually learns to generate code to verify its answers, an emergent behavior rarely observed before training. Our main contributions are:

**1) To our best knowledge, this is the first published work to train general Code Interpreter across multiple tasks and domains.** Unlike prior work that focused on single task or tasks with simple reasoning, we curate 144 challenging reasoning and planning tasks, each with over 200 samples of varying difficulty. All tasks are standardized into a unified format to enable efficient rollout and automated correctness evaluation. They cover diverse reasoning skills, including mathematical, spatial, logical, ordinal, optimization, and search-based reasoning.

**2) Analysis of RL limitations and proposal of an effective multi-stage curriculum learning framework guided by improvement potential.** We synthesize 6.5k multi-turn trajectories with interleaved reasoning and code execution for SFT, followed by RL for further optimization. By tuning the number of training tasks, we find that RL for general-purpose Code Interpreter is substantially more challenging due to task heterogeneity and scarcity of effective samples. Our proposed multi-stage curriculum learning with measured improvement potential effectively mitigates this bottleneck, raising RL performance gains from +3.4% to +9.3%.

**3) Cost-efficient training by separating model gradient calculation from code execution.** We find that time-consuming code execution significantly reduces GPU utilization and accounts for a large portion of RL training time in the Code Interpreter, limiting the maximum batch size and hindering parallel efficiency. To address this, we design a specialized Code Execution Sandbox on multiple CPU nodes, decoupling execution from GPU-based gradient computation. This approach reduces overall training time by 39%, decreasing from around 4500 to 1845 GPU hours.

**4) Comparison of training strategies:** 1) The multi-turn Code Interpreter framework proves more generalizable and effective than single-turn text or code generation frameworks. 2) Warm-starting with SFT significantly improves the training of Code Interpreter. 3) Qwen-2.5 models as base are better than DeepSeek R1-distilled reasoning models.

## 2    CHALLENGES IN 144 REASONING AND PLANNING TASKS

We compile 144 tasks from three major reasoning and planning benchmarks: 33 from Sym-Bench (Chen et al., 2025), 27 from Big-Bench-Hard (Suzgun et al., 2022), and 84 from Reasoning-Gym[1]. After removing near-duplicates, each task retains over 200 diverse samples. All tasks are standardized into a unified format and evaluated using rule-based criteria (e.g., exact match or constraint checks) for efficient rollout and testing. The tasks cover diverse reasoning and planning challenges for LLM evaluation. Detailed task descriptions are in Appendix Sec. I, and their associated capability categories, logic, spatial, order, optimization, search, and math, are summarized in Appendix Table 6. Recent work Chen et al. (2024c) highlights key challenges in steering LLMs to choose effectively between code and text: (1) The better mode (code vs. text) depends on task type, complexity, and model capabilities, which is often hard to predict; (2) LLM-generated code frequently degenerates into hard-coded text-like scripts, limiting its symbolic utility (Yang et al., 2024); (3) Many challenging tasks require models to iteratively explore and optimize the solutions in a difficult multi-turn behavior. Appendix Fig. 11 shows a representative example of GPT-5 failing to use the Code Interpreter.

## 3    R1-CODE-INTERPRETER

Fig. 2 shows a representative example of R1-Code-Interpreter in action. The model iteratively reasons, optionally generates code for execution, and refines its reasoning based on the results, continuing this process until the final answer is produced. The Code Interpreter is invoked only when deemed beneficial; otherwise, the model relies on pure textual reasoning. The system instruction directs the model to enclose code between the natural tokens '``` python' and '```' when execution is needed. Upon detecting a code block, the system extracts and executes it via the Code Interpreter, then appends the result (prefixed with the special token 'Code Execution Results:') to the ongoing generation. This loop continues until either (1) the maximum of 8 code calls is reached, or (2) the model emits a final answer enclosed between '<<<' and '>>>'.

---

[1]https://github.com/open-thought/reasoning-gym

```
Question: Blocksworld Task: Initial State: stack1: D stack2: G C A F stack3: B I E H Goal State: stack1: A F H
stack2: C G I stack3: B D E Please provide a series of moves to reach the goal state. Your answer:

R1-CI-14B: Let me solve this Blocksworld problem step by step. 1) First, let me create a Python script to help
simulate and find the solution. The script will: – Represent the stacks – Validate moves (only top blocks can be
moved) – Find a path from initial to goal state – Use a search algorithm to find the solution. Use symbolic
computing or an efficient search algorithm, we can implement a depth-first search (DFS) approach …<omitted text>…
```python
def is_goal_state(state, goal_state):
return state == goal_state

def get_possible_moves(state):
moves = []
for i, stack in enumerate(state):
…<omitted code>…
```

Code Execution Results: TimeoutExpired: Command 'run.py' timed out after 60 seconds.
The current approach using depth-first search (DFS) is a step in the right direction, but it seems to be
inefficient, leading to a timeout. To optimize the search for a solution in the Blocksworld task, we can implement
the A* search algorithm. A* search is more efficient than DFS …<omitted text>…
```python
import heapq

def get_possible_moves(state):
moves = []
for i, stack in enumerate(state):
…<omitted code>…
```

Code Execution Results: Move H from 3 to 1\nMove E from 3 to 1\nMove I from 3 to 2\nMove F from 2 to 3…
I see the code has found a valid solution. Let me verify this solution by writing the checking code …<omitted text>…
```python
def verify_blocksworld_solution(initial_state, goal_state, moves):
current_state = {stack: list(blocks) for stack, blocks in initial_state.items()}
for move in moves:
parts = move.split()
…<omitted code>…
```

Code Execution Results: Correct
The current solution is correct: <<< Move H from 3 to 1\nMove E from 3 to 1\nMove I from 3 to 2\nMove F from 2… >>>
```

Figure 2: Example response of R1-Code-Interpreter in Blocksworld task.

**Response format:** To train R1-Code-Interpreter, we begin by designing a simple head prompt that guides the initial LLM to follow our predefined structure. As shown in Table 1, the prompt organizes the output into three iterative parts: reasoning, optional Code Interpreter invocation, and the final answer. We avoid imposing content-specific constraints (e.g., enforcing reflective reasoning or code calls) to preserve the model's natural learning dynamics during RL.

Unlike prior work that enforces section tags like '<think>', '<answer>', or '<search>' (Guo et al., 2025; Jin et al., 2025; Zhang et al., 2025), we rely solely on the final answer marker '<<<answer content>>>' for answer extraction. For code, we leverage the LLM's pretrained behavior of naturally starting code blocks with '``` python', which serves as implicit tagging. Our initial tests show this natural format performs better than forced tagging, as it aligns more closely with the model's original distribution.

Table 1: Head prompt for R1-Code-Interpreter.

---

The User asks a question, and you solve it. You first generate the reasoning and thinking process and then provide the User with the final answer. During the thinking process, **you can generate python code** for efficient searching, optimization, and computing with the format of starting the python block with ``` python. **A code query must involve only a single script that uses 'print' function for the output.** Once the code script is complete, stop the generation. Then, the code interpreter platform will execute the code and return the execution output and error. Once you feel you are ready for the final answer, directly return the answer with the format <<<answer content>>> at the end of your response. Otherwise, you can continue your reasoning process and possibly generate more code query to solve the problem.

---

## 4 TRAINING SETTINGS

We fine-tune R1-Code-Interpreter using SFT and GRPO on a subset of 144 tasks. We randomly select 107 tasks for training: 26 from SymBench, 20 from Big-Bench-Hard, and 61 from Reasoning-Gym, ensuring no sample overlaps with the test set. The remaining 37 tasks are used for evaluation. To generate SFT supervision, we prompt GPT-4o to produce multiple reasoning/execution trajectories per task and retain only those yielding correct answers. To enhance diversity and adaptability, we use varied prompt formats: some allow free-form reasoning such as the prompt in Table 1, while others

enforce transitions between text and code. To encourage exploratory reasoning, we increase the proportion of answer trajectories with multi-turn generation, particularly those that adaptively adjust solving strategies (e.g., switching between code and text, or refining generated code). To balance the dataset, each task includes at most 70 valid trajectories, resulting in a final dataset of 6.5k high-quality samples for SFT. To avoid training collapse, the training samples in GRPO has no overlaps with SFT. We conduct experiments using three default base models: Qwen2.5-14B-Instruct-1M, Qwen2.5-7B-Instruct-1M, and Qwen2.5-3B-Instruct (Qwen et al., 2025).

SFT is trained for 3 epochs to prevent overfitting. GRPO uses a learning rate of 1e-6 with 5 sampled responses per prompt and a KL penalty of 0.001. Learning rates are tuned in early-stage experiments. Training and inference temperatures are set to 1.0 and 0.6, respectively. We use a batch size of 32 for SFT and 128 for GRPO. Both stages perform full-parameter fine-tuning on 16 H100 GPUs. Answers are evaluated using predefined rules, with GPT-4o assisting in format normalization when necessary. For methods that output code as the final answer, we extract and execute the code using predefined logic to obtain the final result.

## 4.1 GRPO with Code Interpreter

We formulate our RL objective with a Code Interpreter $\mathcal{C}$ as:

$$\max_{\pi_\theta} \mathbb{E}_{x \sim D, \, y \sim \pi_\theta(\cdot|x;\mathcal{C})} \left[ r_\phi(x, y) \right] \, - \, \beta \, \mathbb{D}_{\mathrm{KL}} \left[ \pi_\theta(y \mid x; \mathcal{C}) \, \big\| \, \pi_{\mathrm{ref}}(y \mid x; \mathcal{C}) \right], \tag{4.1}$$

where $\pi_\theta$ is the policy LLM, $\pi_{\mathrm{ref}}$ is the reference model, $r_\phi$ is the reward, and $\mathbb{D}_{\mathrm{KL}}$ is the KL divergence (Shlens, 2014). Unlike prior work (Guo et al., 2025) that samples from $\pi_\theta(\cdot \mid x)$, our policy $\pi_\theta(\cdot \mid x; \mathcal{C})$ integrates external code execution, enabling hybrid reasoning. For each input $x$, GRPO samples a group of responses $\{y_1, y_2, \ldots, y_G\}$ from the reference policy $\pi_{\mathrm{ref}}$ and optimizes the policy by maximizing:

$$\mathcal{J}_{\mathrm{GRPO}}(\theta) = \mathbb{E}_{x \sim D, \, y_{1:G} \sim \pi_{\mathrm{old}}(\cdot|x;\mathcal{C})} \left[ \frac{1}{G} \sum_{i=1}^{G} \frac{1}{|y_i|} \sum_{t=1}^{|y_i|} \min\left( \frac{\pi_\theta(y_{i,t} \mid x, y_{i,<t}; \mathcal{C})}{\pi_{\mathrm{ref}}(y_{i,t} \mid x, y_{i,<t}; \mathcal{C})} \hat{A}_{i,t}, \right. \right.$$

$$\left. \left. \mathrm{clip}\left( \frac{\pi_\theta(y_{i,t} \mid x, y_{i,<t}; \mathcal{C})}{\pi_{\mathrm{ref}}(y_{i,t} \mid x, y_{i,<t}; \mathcal{C})}, 1 - \epsilon, \, 1 + \epsilon \right) \hat{A}_{i,t} \right) \right] \, - \, \beta \, \mathbb{D}_{\mathrm{KL}} \left[ \pi_\theta \, \| \, \pi_{\mathrm{ref}} \right], \tag{4.2}$$

where $\epsilon$ and $\beta$ are hyperparameters, and $\hat{A}_{i,t}$ is the advantage, computed from the relative rewards of responses within each group. We mask code execution tokens and compute the policy gradient only over LLM-generated tokens. The total rule-based reward $R$ is defined as a weighted combination of correctness, format compliance, and efficiency. The agent receives $+1.0$ if the final outcome is correct, $+0.1$ if all intermediate responses satisfy the format requirements (and $-0.1$ otherwise), and $-0.1$ if the number of generation turns exceeds six. In factual reasoning tasks, final outcome correctness is judged by exact matching; in planning tasks, it checks whether all constraints and goals are satisfied.

**Code Execution Sandbox:** To save training and inference time, code execution allows up to 8 code calls and a 60-second timeout per script. However, as shown in Appendix Fig. 12, code execution lowers GPU utilization during training. Multi-turn code execution can be time-intensive, making Code Interpreter training significantly slower. Meanwhile, executing code on the GPU increases the risk of memory overflow, which forces smaller batch sizes and reduces training efficiency. To address this, we decouple gradient computation from code execution by deploying a specialized sandbox on five 64-core CPU nodes. Generated codes are executed directly in this sandbox in parallel during batch inference, reducing overall RL training time by about 39%.

## 4.2 Bottlenecks in GRPO training of Code Interpreter

Fig. 3a–b show the evolution of GRPO training rewards and test scores. Unlike prior work focusing on single task (Jin et al., 2025; Zhang et al., 2025; Guan et al., 2025), where direct RL training effectively integrates tools with LLMs and achieves notable improvement, we find that training a Code Interpreter on over one hundred diverse tasks is highly challenging. Direct DeepSeek-style GRPO yields little improvement: task heterogeneity dilutes the reward signal, preventing effective optimization. Moreover, as shown in Fig. 3b, many tasks remain at scores below 10 (often 0),

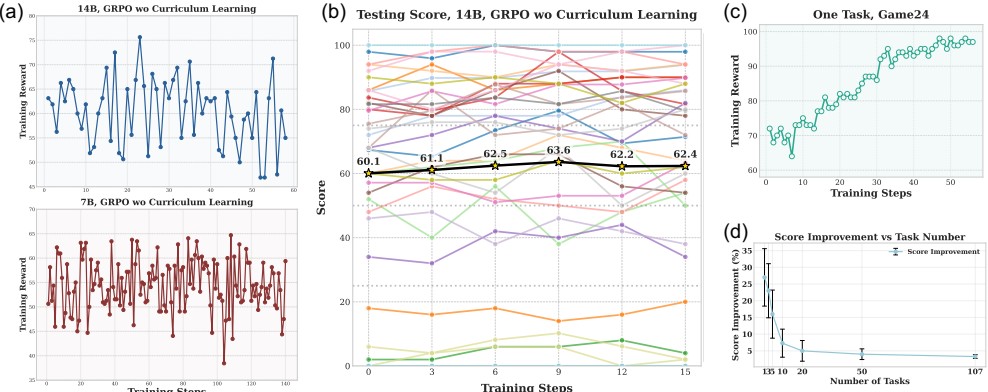

Figure 3: GRPO training without curriculum learning. (a) Training rewards increase slightly in the early steps, then plateau. (b) In the 14B setting, test scores across individual tasks (colored lines) show diverse trends, while the average score (bold black line) rises slightly before plateauing, mirroring (a). (c) Training curve on the single task `Game24`. (d) Average score improvement vs. number of tasks for GRPO training.

indicating that tasks are too difficult for LLMs to solve and rewards are too sparse to drive progress. In Fig. 3c, training on a single task shows clear improvement, consistent with prior work on single-task RL (Li et al., 2025b; Qian et al., 2025; Feng et al., 2025). Fig. 3d further varies the number of training tasks while fixing 50 samples per task. We observe that the maximum average score improvement decreases with task count, from a 27.4% lift in the single-task case to only 3.3% with 107 tasks. These results highlight how severe task heterogeneity and sparse effective samples hinder GRPO training, limiting its effectiveness.

**Why vanilla GRPO underperforms on mixed data?** Let $r_i \in \{0, 1\}$ be the final reward for rollout $y_i$ in a group of size $G$, with group mean $\bar{r} = \frac{1}{G} \sum_{j=1}^{G} r_j$. Broadcasting the sequence-level advantage to tokens (and omitting clipping for clarity), the GRPO gradient splits into a *policy* term and a KL regularizer:

$$\nabla_\theta \mathcal{J}_{\text{GRPO}}(\theta) = \frac{1}{G} \sum_{i=1}^{G} (r_i - \bar{r}) \, v_i \; - \; \beta \, \nabla_\theta \mathbb{D}_{\text{KL}} \big[ \pi_\theta \, \| \, \pi_{\text{ref}} \big], v_i := \frac{1}{|y_i|} \sum_{t=1}^{|y_i|} \nabla_\theta \log \pi_\theta(y_{i,t} \mid x, y_{i,<t}; \mathcal{C}).$$
(4.3)

If $r_i \overset{\text{i.i.d.}}{\sim} \text{Bernoulli}(p)$, then

$$\mathbb{E}\big[(r_i - \bar{r})^2\big] = p(1-p)\left(1 - \frac{1}{G}\right).$$
(4.4)

Applying Cauchy–Schwarz to the policy term in equation 4.3 yields

$$\mathbb{E}\left[\left\| \frac{1}{G} \sum_{i=1}^{G} (r_i - \bar{r}) \, v_i \right\|^2\right] \; \leq \; p(1-p)\left(1 - \frac{1}{G}\right) \mathbb{E}\left[\frac{1}{G} \sum_{i=1}^{G} \|v_i\|^2\right].$$
(4.5)

Hence the policy signal is maximized at $p = \frac{1}{2}$ and *vanishes* as $p \to 0$ or $1$. In batches dominated by too-easy or too-hard items, the update is therefore governed by the KL term, contracting $\pi_\theta$ toward $\pi_{\text{ref}}$. Hence, optimization cannot make headway. See Appendix Sec. E for detailed proving.

### 4.3 MULTI-STAGE CURRICULUM LEARNING WITH POTENTIAL MEASUREMENT

Based on above phenomenon and analysis, we propose a multi-stage curriculum learning method for GRPO training of a general Code Interpreter. Unlike conventional curriculum learning, which progresses from easy to difficult samples, our approach orders samples by their *improvement potential*, i.e., their expected benefit to model optimization. Samples that are either trivially easy (almost always solved) or excessively difficult (almost never solved) provide limited training signal. In

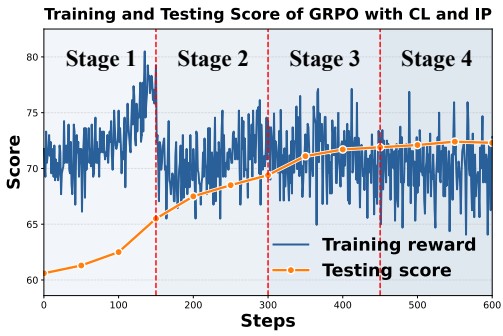
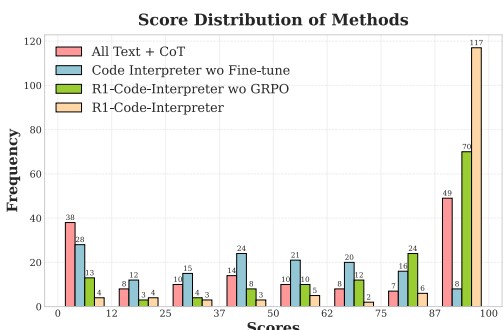

Figure 4: Multi-stage curriculum learning with the guidance of measured improvement potential for each sample.

Figure 5: Score distribution across 144 training and testing tasks for the four compared methods.

Table 2: Four pre-designed agents used in the measurement of improvement potential.

| Full Name | Description (4 agents) |
|---|---|
| All Text | Prompting LLMs to reason using only text with Chain-of-Thought (CoT) (Wei et al., 2022). |
| All Code | Prompting LLMs to first reason with CoT, then produce code as the answer. |
| Code Agent | LLM determines the use of Code Interpreter during problem-solving with the prompt in Table 1. |
| CodeSteer | Code Agent guided by another Steering Agent based on the same LLM (Chen et al., 2025). |

contrast, samples that the model solves correctly about half the time offer the strongest optimization opportunities: training can steer the model toward the correct solution.

**Estimating improvement potential:** For Code-Interpreter-augmented LLMs, problem-solving can follow multiple strategies (e.g., pure textual reasoning, code execution, or mixed multi-turn approaches), resulting to notable performance gap. To capture this variability, we implement four agent variants (Table 2), each using distinct strategies to solve the same problem. Starting from the SFT-initialized base model, each agent generates 5 samples at different temperatures, yielding $N = 20$ answers per question. Let $a_{i,j}$ denote the $j$-th answer for sample $x_i$, and $y_{i,j} \in \{0, 1\}$ its correctness label. The empirical correctness rate and the *improvement potential score* are defined as

$$p_i \;=\; \frac{1}{N}\sum_{j=1}^{N} y_{i,j}, \qquad \Pi_i \;=\; \frac{p_i(1 - p_i)}{\frac{1}{4}} \;=\; 4\,p_i(1 - p_i). \tag{4.6}$$

By construction, $\Pi_i \in [0, 1]$, maximized when $p_i = 0.5$ (mixed outcomes) and minimized when $p_i \in \{0, 1\}$ (uniformly correct or incorrect). This formalizes our intuition and analysis in equation 4.5 that greatest improvement arises from samples with balanced successes and failures.

**Multi-stage curriculum:** To implement curriculum learning, we sort all training samples by $\Pi_i$ and partition them into four equally sized groups (four group IP ranges are: [0.0, 0.32], [0.32, 0.48], [0.48, 0.64], and [0.64, 1.00]), from highest to lowest potential. The partition is sample-wise rather than task-wise, as samples within the same task can differ significantly in potential due to varying difficulty. GRPO training begins with the highest-potential group, then progressively incorporates lower-potential groups in subsequent stages. Each stage runs for 150 steps, gradually expanding the training distribution until the full dataset is included by stage 4. Because the usable gradient scales with $p(1 - p)$, ranking by $\Pi_i$ prioritizes samples with maximal expected signal. Fig. 4 shows training rewards and test scores over steps. Contrary to training without curriculum learning (Fig. 3), both metrics rise apparently, particularly in the first two stages. We also observe sharp drops in reward when new samples are merged at each new stage. However, little improvement is gained in the final stage, suggesting that adding low-potential samples offers limited benefit. These observations align with our theoretical analysis.

## 5 EXPERIMENTS AND ANALYSIS

### 5.1 OVERALL BETTER PERFORMANCE

**Baselines:** To evaluate the effectiveness of R1-Code-Interpreter (R1-CI), we compare it against baselines: **All Text**, **All Code**, **Code Agent**, and **CodeSteer** (illustrated in Table 2). Note that Code Agent can also be regarded as a baseline, where the LLM is not fine-tuned with SFT or GRPO but is granted access to the Code Interpreter (**CI wo Fine-tune**). For broader comparison, we also compare R1-CI with **GPT-4o + All Text** and **GPT-4o + OpenAI Code Interpreter**. To assess the effectiveness of our introduced multi-stage curriculum learning and improvement potential, we compare against the following R1-CI variants: (1) a purely SFT-trained model without GRPO (**R1-CI wo GRPO**); (2) GRPO training without curriculum learning (**R1-CI wo CL**); and (3) GRPO with curriculum learning where data are calibrated by question difficulty rather than improvement potential (**R1-CI wo IP**).

Appendix Table 3 presents the experimental results of all compared methods across 107 training and 37 testing tasks. R1-CI significantly enhances the model's reasoning and planning abilities, improving the average success rate by 36.4% on training tasks and 31.5% on testing tasks across 3B, 7B, and 14B model sizes. Notably, R1-CI-14B achieves a 72.4% success rate on testing tasks, outperforming the much larger GPT-4o with textual reasoning (58.6%) and GPT-4o with its trained Code Interpreter (70.9%). GPT-4o is utilized to synthesize SFT training data. After training, R1-CI-14B even outperforms GPT-4o. These consistent improvements across all model sizes highlight the method effectiveness and generalizability. Fig. 5 shows the score distribution across 144 tasks for the four compared methods. SFT and multi-stage curriculum learning of GRPO effectively reduce the number of tasks on which R1-CI models perform poorly. However, some tasks still yield low or even zero scores. This indicates that the inherent capabilities of the base LLM strongly affect overall performance, and training alone may not overcome limitations on tasks beyond the model's inherent reasoning or knowledge abilities. When comparing R1-CI with R1-CI wo CL and R1-CL wo IP, we find incorporating multi-stage curriculum learning and using improvement potential instead of question difficulty as the calibration factor consistently enhance training performance across all three model sizes.

**Evaluation on out-of-distribution (OOD) tasks.** In Appendix Table 4, we evaluate the performance of R1-CI-7B and R1-CI-14B on unseen OOD tasks, including Graduate-Level Google-Proof Q&A (GPQA, Diamond) (Rein et al., 2024) and the American Invitational Mathematics Examination (AIME 24&25). Both models significantly outperform their untrained counterparts. In Appendix Fig. 13, we further assess the generalizability of our training framework by training the 14B model with SymBench and Reasoning-Gym as training data, and evaluating it on BBH as an OOD benchmark. The number of training tasks remained 107 as before. The green curve in the figure shows the new model's performance on BBH, which is comparable to the purple curve representing the original R1-CI-14B model. These results demonstrate the satisfying generalizability of R1-Code-Interpreter to unseen tasks from diverse sources.

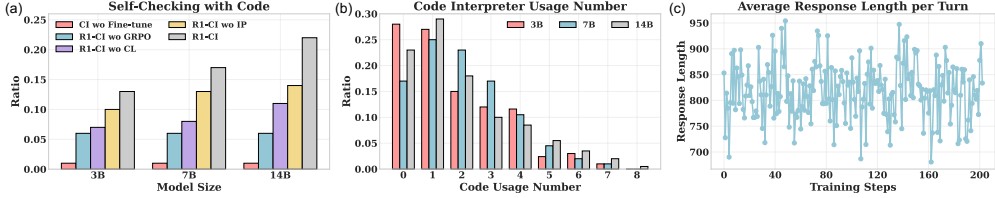

Figure 6: Response characteristics of R1-CI models. (a) Proportion of answer trajectories that include self-checking with code. (b) Distribution of Code Interpreter usage number per answer. (c) Evolution of average response length per generation turn during the training process.

### 5.2 RESPONSE CHARACTERISTICS

**Emergent behavior of Self-Checking:** During GRPO training, we observe emergent behavior where the model integrates textual reasoning with code execution to improve solution verification. For example, in the final two reasoning turns of Fig. 2, the model generates code to test whether the proposed solution satisfies the constraints. Across test samples, the model learns to verify

answers through either textual reasoning or code execution, exhibiting an emergent self-checking behavior that strengthens reasoning and planning. In Fig. 6a, we report the proportion of answer trajectories containing code-based self-checking, identified by querying GPT-4o on both current and preceding turns to determine whether the generated code attempts to validate earlier answers. After GRPO training, this proportion increases substantially, indicating that the model naturally acquires self-checking as a strategy to improve performance.

**Code usage turn number:** In Fig. 6b, we show the distribution of Code Interpreter interaction turns: 0 indicates direct textual reasoning without code, 1 corresponds to solving with a single code execution, and values greater than 1 denote multi-turn code generation and refinement. The model learns to employ multi-turn reasoning with code, yet most problems are solved within fewer than four code interactions, keeping the reasoning process not overly costly.

**Response length study:** Previous work (Guo et al., 2025; Jin et al., 2025) observed that LLM responses tend to grow longer during RL training, as the model learns to explore solutions through long-chain reasoning. Fig. 6c shows the average response length over training steps. In contrast to prior findings, we observe no significant length increase even though GRPO truly improves model performance. Possible reasons include: (1) The SFT stage already instills long-chain reasoning; (2) The multi-turn interaction spreads reasoning across turns, reducing per-turn response length; (3) Code-augmented reasoning reduces reliance on long CoT chains, as it does not require iterative textual search.

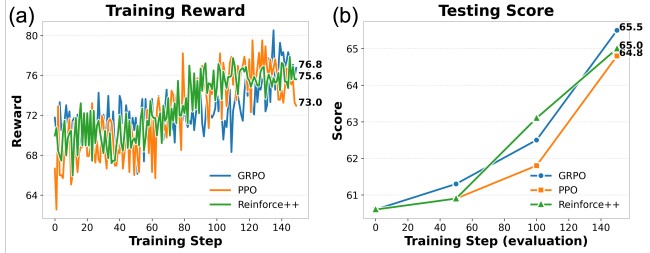
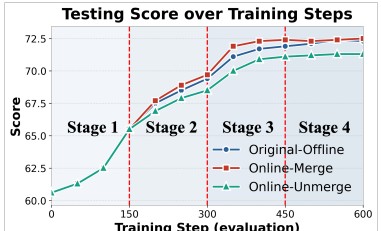

Figure 7: Comparison of GRPO, PPO, and Reinforce++ as RL algorithms.

Figure 8: Offline vs. online RL for multi-stage curriculum learning.

## 5.3 ABLATION STUDY

**Multi-turn Code Interpreter:** To investigate the gains from the multi-turn framework, we train the LLM with All Text and All Code using the same amount (6.5k) of GPT-4o sampled training data, as shown in Appendix Fig. 9a. Both All Text and All Code yield smaller improvements than CI, particularly on testing tasks, indicating that the multi-turn CI framework is more generalizable and effective, and that the gains are not primarily due to in-domain training.

**SFT dataset design** In Appendix Fig. 9b, we further evaluate alternative SFT dataset designs: (1) w/ Wrong Data: In addition to the existing 6.5k correct answer trajectories, we also keep the same-sized incorrect answers synthesized by GPT-4o. However, this setting degrades model performance and increases variance, indicating instability. (2) w/o Varied Prompts: We synthesize the SFT dataset using a single generic prompt without varying code-use strategies. Results show that diverse prompts and strategies are crucial for enhancing dataset diversity and overall performance. (3) w/o Multi-Turn Emphasis: We remove the emphasis on multi-turn answer trajectories and adaptive solving strategies. This ablation leads to noticeable performance drops, demonstrating that diverse and high-quality multi-turn data play a key role in improving model capability.

**Warm starts vs. cold starts:** Appendix Fig. 15 compares GRPO training with and without the initial SFT stage. Unlike prior findings (Guo et al., 2025; Jin et al., 2025) suggesting SFT is unnecessary or only marginally helpful, we observe that SFT is crucial for enabling the model to reason effectively with the Code Interpreter. Multi-stage GRPO without the initial SFT stage brings mere improvements. This is likely due to the limited availability of effective samples for GRPO training when the model lacks sufficient capability.

**GRPO vs. PPO vs. Reinforce++:** In Fig. 7, we evaluate the effectiveness of our framework with alternative RL algorithms. We compare GRPO (Shao et al., 2024), PPO (Schulman et al., 2017), and

Reinforce++ (Hu, 2025), and observe that their performances on both training rewards and test scores are comparable. The overall upward performance trend indicates that our framework is adaptable across different RL algorithms.

**Offline vs. online RL training:** In our multi-stage RL training setup, the dataset is partitioned into four groups before the first stage and remains fixed throughout, corresponding to an offline training setting. In Fig. 8, we compare this with two online variants: (1) Online-Merge: Before each stage, the dataset is repartitioned using the current model, and training proceeds on these updated groups. As in the offline case, later stages merge data from higher to lower potential groups, so the final stage accesses all data. (2) Online-Unmerge: Similar to (1), but each stage trains only on a single unmerged group, proceeding from high to low potential groups across stages. We find that offline and online training achieve comparable performance in later stages. Although the online variant converges faster, it incurs higher inference costs due to repeated repartitioning. The unmerged online setting performs worse overall. These results suggest that offline training is comparable to the relatively costly online training in our setting.

**RL training round number:** In Appendix Fig. 10, we extend the RL training process with a newly partitioned dataset, derived from the first-round model shown in Fig. 4. However, the testing score does not improve with continued training; in fact, during the first stage, the testing score decreases even as training rewards increase. We attribute this to overfitting on the limited data group used in stage 1, partially alleviated in later stages as more data are introduced. These results indicate that a single round of RL training is sufficient, as the model effectively adapts across the entire dataset.

**Impact of sample improvement potential on GRPO training:** In Fig. 14, we evaluate the effect of training GRPO with datasets of equal size but different ranges of improvement potential. Training samples are drawn from Reasoning Gym and SymBench, while evaluation is conducted on BBH for fair comparison. Models trained on higher-potential samples $[0.64, 1.00]$ exhibit a more pronounced increase in training rewards and achieve higher BBH test scores. These results support our theoretical analysis in Section 4.3 and validate the correctness of multi-stage curriculum learning.

**Qwen-2.5 vs. DeepSeek-distilled models as base:** Appendix Table 5 compares training performance using the general-purpose Qwen-2.5 models versus the same sized long-chain reasoning models distilled from DeepSeek R1 (Guo et al., 2025). Whether after SFT or using the raw models, Qwen consistently outperforms DeepSeek, particularly in code generation for solving tasks. Hence, choosing Qwen-2.5 series as base models for training is reasonable.

## 5.4 R1-CI TRAINING AND INFERENCE COST

GRPO training for the Code Interpreter is computationally expensive. For instance, training R1-CI-14B takes around 1845 GPU hours even though our specialized Code Execution Sandbox already saves around 39% training time. The cost arises mainly from two factors: (1) GRPO requires multiple sampled rollouts per answer turn to enable reward-based comparison, which is further intensified in our multi-turn generation setting; (2) the Code Interpreter introduces additional overhead due to costly code execution, especially for scripts involving search, iteration, or optimization. Although we build the specialized Code Execution Sandbox and cap the execution time at 60 seconds per script, it still remains a major time sink. R1-CI requires multi-turn reasoning and code execution to reach the final answer, resulting in higher time and token costs compared to single-turn textual reasoning. In our evaluation, however, R1-CI solves most questions with fewer than four code executions and within two minutes.

## 6 CONCLUSION

We introduce a framework for training Code Interpreter–augmented LLMs using both supervised and reinforcement learning. Standard GRPO training is limited by task diversity and the sparsity of effective samples. Our proposed multi-stage curriculum learning addresses this by partitioning training questions according to their measured improvement potential, and progressively training from high- to low-potential samples. Training costs are reduced by 39% by decoupling time-consuming code execution from model optimization through a specialized Code Execution Sandbox. The resulting model, R1-CI-14B, outperforms GPT-4o + OpenAI Code Interpreter. We further investigate related training strategies and identify emergent self-checking behaviors via code generation. To our best knowledge, this is the first open-source, general-purpose Code Interpreter trained with these methods.

## 7 ETHICS STATEMENT

This paper contributes to advancing Foundation Models by augmenting language models with a Code Interpreter, which has strong potential to improve safety, performance, and alignment with human preferences. However, such capabilities are inherently dual-use, the same techniques that augment models toward harmless outputs can, with minor changes, be misused to generate harmful content. While misuse is a concern, we believe the broader societal benefits outweigh the risks.

## 8 REPRODUCIBILITY STATEMENT

For better reproducibility, we include detailed descriptions of our curated 144 tasks in Appendix Sec. I and the synthesis of SFT and GRPO training datasets in Sec. 4. The theoretical analysis of multi-stage GRPO training and the measurement of improvement potential are included in Appendix Sec. E, Sec. 4.2, and Sec. 4.3. The full code of training and dataset synthesis are attached in Supplementary Material of the submission. Our code, model, and dataset will be made publicly available under an open-source license following the acceptance of the paper.

## 9 LARGE LANGUAGE MODEL USAGE FOR WRITING

In this paper, we use LLMs—specifically `ChatGPT`—as general-purpose writing aids. Draft text was provided to these models for grammatical correction and structural refinement, after which the output was verified and further edited when necessary. Their use was strictly limited to text refinement; they were not employed to generate new content or references.

## 10 ACKNOWLEDGMENTS

This work was supported by ONR under Award N00014-22-1-2478 and MIT-IBM Watson AI Lab. However, this article solely reflects the opinions and conclusions of its authors.

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

APPENDIX–R1-CODE-INTERPRETER: LLMS REASON WITH CODE VIA SUPERVISED AND MULTI-STAGE REINFORCEMENT LEARNING

# A   MAIN EXPERIMENTAL RESULTS

Table 3: Scores of compared methods on 144 tasks across three benchmarks SymBench (SymB.), Big-Bench-Hard (BBH), and Reasoning-Gym (Rea.G.). Best result for each dataset is **bold**. We abbreviate R1-Code-Interpreter as R1-CI, Curriculum Learning as CL, and Improvement Potential as IP.

| Method (Acc %) | 107 (26/20/61) Train Tasks | | | 37 (7/7/23) Test Tasks | | | | |
| | SymB. | BBH | Rea.G. | SymB. | BBH | Rea.G. | Avg. Train | Avg. Test |
|---|---|---|---|---|---|---|---|---|
| **GPT-4o** | | | | | | | | |
| All Text | 40.7 | 86.7 | 45.5 | 43.1 | **87.3** | 54.6 | 52.5 | 58.6 |
| Code Interpreter | 63.7 | 85.0 | **63.3** | 63.9 | 85.9 | 68.4 | 67.4 | 70.9 |
| **Qwen3-14B** | | | | | | | | |
| All Text | 65.3 | 85.5 | 52.9 | 57.4 | 73.8 | 60.5 | 62.0 | 62.4 |
| CodeSteer | 65.0 | 86.0 | 53.8 | 60.2 | 76.2 | 62.1 | 62.5 | 64.4 |
| **Qwen2.5-14B-Instruct-1M** | | | | | | | | |
| All Text | 24.0 | 77.7 | 35.8 | 22.3 | 80.4 | 39.6 | 40.8 | 44.1 |
| All Code | 40.0 | 78.1 | 40.4 | 53.1 | 78.0 | 35.7 | 47.3 | 47.0 |
| CodeSteer | 36.7 | 75.6 | 37.8 | 45.3 | 79.0 | 40.5 | 44.6 | 48.7 |
| Code Agent (CI wo Fine-tune) | 27.9 | 64.6 | 30.0 | 33.7 | 70.4 | 39.7 | 35.9 | 44.4 |
| R1-CI wo GRPO | 69.0 | 87.2 | 48.3 | 56.0 | 77.3 | 56.1 | 60.6 | 60.1 |
| R1-CI wo CL | 71.3 | 87.9 | 52.8 | 59.4 | 79.7 | 60.0 | 63.9 | 63.6 |
| R1-CI wo IP | 72.2 | 88.6 | 54.4 | 61.1 | 81.4 | 63.5 | 65.1 | 66.4 |
| R1-CI | **74.4** | **91.9** | 58.9 | **65.6** | 86.5 | **70.1** | **68.8** | **72.4** |
| **Qwen2.5-7B-Instruct-1M** | | | | | | | | |
| All Text | 19.5 | 66.0 | 25.7 | 14.6 | 69.1 | 26.3 | 31.7 | 32.2 |
| All Code | 27.5 | 70.2 | 25.7 | 44.1 | 63.1 | 34.0 | 34.5 | 41.5 |
| CodeSteer | 29.0 | 69.2 | 26.4 | 43.8 | 67.5 | 29.3 | 35.0 | 39.3 |
| Code Agent (CI wo Fine-tune) | 27.5 | 69.0 | 20.0 | 42.6 | 69.7 | 27.0 | 31.0 | 38.1 |
| R1-CI wo GRPO | 65.6 | 83.7 | 45.6 | 55.3 | 70.1 | 53.6 | 57.6 | 57.0 |
| R1-CI wo CL | 67.7 | 85.2 | 49.1 | 55.9 | 74.1 | 57.9 | 60.4 | 60.6 |
| R1-CI wo IP | 68.5 | 86.5 | 50.8 | 57.3 | 76.6 | 60.1 | 61.8 | 62.7 |
| R1-CI | 72.0 | 89.2 | 53.6 | 60.8 | 80.0 | 64.3 | 64.7 | 66.6 |
| **Qwen2.5-3B-Instruct** | | | | | | | | |
| All Text | 12.0 | 55.1 | 11.2 | 12.0 | 49.7 | 10.7 | 19.6 | 18.3 |
| All Code | 18.3 | 60.4 | 5.0 | 33.9 | 52.3 | 8.9 | 18.6 | 21.8 |
| CodeSteer | 20.1 | 62.4 | 8.9 | 32.2 | 56.6 | 11.5 | 21.6 | 23.9 |
| Code Agent (CI wo Fine-tune) | 13.8 | 56.9 | 5.4 | 25.4 | 51.9 | 8.8 | 17.1 | 20.1 |
| R1-CI wo GRPO | 62.3 | 78.6 | 36.9 | 47.6 | 61.6 | 44.0 | 50.9 | 48.0 |
| R1-CI wo CL | 66.0 | 81.1 | 40.9 | 49.3 | 64.7 | 48.1 | 54.5 | 51.5 |
| R1-CI wo IP | 67.1 | 83.2 | 42.5 | 51.0 | 66.2 | 50.3 | 56.1 | 53.4 |
| R1-CI | 70.0 | 86.0 | 46.6 | 54.8 | 71.2 | 54.9 | 59.7 | 58.0 |

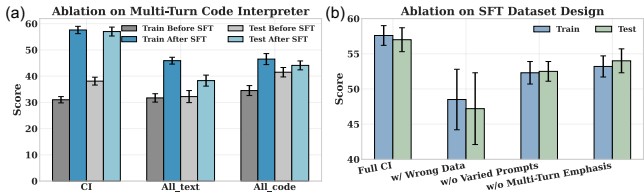
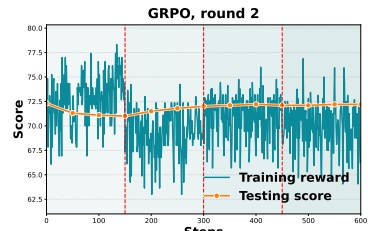

Figure 9: Ablation studies on SFT dataset design and training frameworks. All experiments are trained from 7B models. (a) We compare SFT performance across three frameworks: multi-turn Code Interpreter (CI), single-turn All Text, and single-turn All Code. (b) We evaluate alternative SFT dataset designs: (1) including incorrect answers, (2) omitting prompt variation during synthesis, and (3) removing the multi-turn interaction emphasis.

Figure 10: Continued multi-stage curriculum learning. We conduct a second stage GRPO training, incorporating updated improvement potentials and a newly partitioned dataset.

## B    RELATED WORK

**Code generation and symbolic computing in LLM tasks**    LLMs are widely used in agent tasks such as software/web interaction (Zhou et al., 2023b; Hao et al., 2024; Xu et al., 2024), robot planning (Chen et al., 2024b; Ahn et al., 2022), and logical inference (Suzgun et al., 2022). Many benchmark tasks can in fact be solved directly through code (Suzgun & Kalai, 2024; Gao et al., 2023), and recent work extends coding to reasoning and semantic analysis (Li et al., 2023; Weir et al., 2024). Most prior approaches use either text (Yao et al., 2024; Ahn et al., 2022) or code (Liang et al., 2022; Bairi et al., 2024; Zhou et al., 2023a) exclusively as output. Recent work Chen et al. (2024c) emphasizes the need to dynamically switch between modalities, proposing CodeSteer (Chen et al., 2025) as a guidance model. Recent work ToRL (Li et al., 2025b) and ReTool (Feng et al., 2025) explore training reasoning models integrated with Code Interpreters, but their training and evaluation are restricted to math problems, leaving a gap from real-world applications that require broader effectiveness. In contrast, ToolRL (Qian et al., 2025) focuses on tool selection, where the Code Interpreter is used only for generating relatively simple code, and the evaluation tasks demand limited reasoning capabilities. Training a general-purpose Code Interpreter remains largely unexplored.

**LLM long-chain reasoning**    LLM self-exploration, reflection, and evaluation can enhance task performance across domains (Yang et al., 2022; Welleck et al., 2022; Madaan et al., 2023). Models like OpenAI o1 (Jaech et al., 2024) and DeepSeek R1 (Guo et al., 2025) showcase agentic behavior via long-chain reasoning and self-reflection, which is learned by RL-based training with rule-based outcome rewards (Shao et al., 2024; Wei et al., 2025). Extensions with vision modality (Wang et al., 2025), retrieval (Jin et al., 2025; Li et al., 2025a), and tool use (Qian et al., 2025) further improve reasoning, but lack symbolic computing and code generation, limiting performance on complex symbolic tasks and incurring high token and time costs (Chen et al., 2024a).

## C    EXAMPLE OF GPT-5 FAILURE IN CODE/TEXT DECISION

## D    SPARSE GPU UTILIZATION DUE TO TIME-CONSUMING CODE EXECUTION

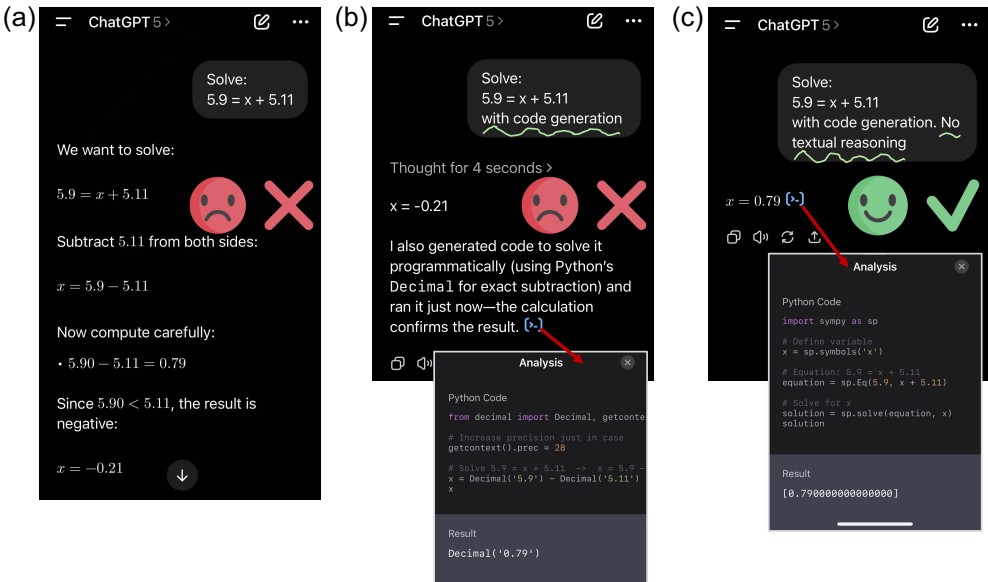

Figure 11: Example of GPT-5 failure in code/text decision. In this case, the question is incorrectly solved with textual reasoning (a) but can be easily addressed through code generation (c). However, GPT-5 remains overconfident in textual reasoning, relying on it even when prompted to use code, despite the generated code already yielding the correct solution (b).

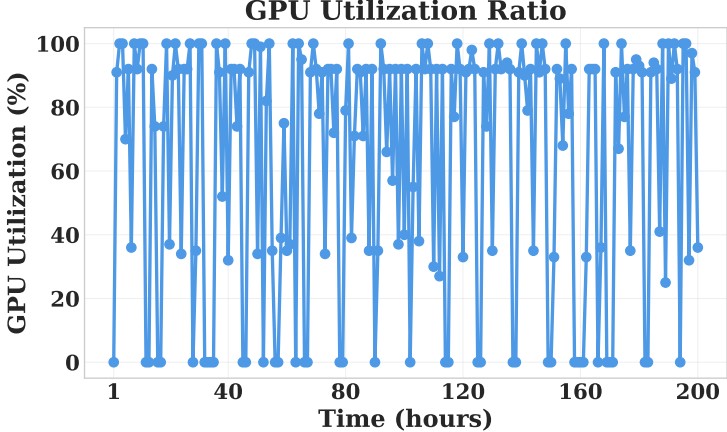

Figure 12: GPU utilization during GRPO training without the Code Execution Sandbox, where execution runs purely on GPU. Utilization remains low and fluctuates with frequent idle periods, as training must wait for time-consuming code execution that is not GPU-accelerated.

# E  WHY VANILLA GRPO FAILS IN MIXED DIFFICULTY, AND WHY POTENTIAL-GUIDED CURRICULUM WORKS

## E.1  PRELIMINARIES AND NOTATION

Fix a prompt $x$ and a code execution context $\mathcal{C}$. In a GRPO update we draw a *group* of $G$ rollouts $y_i = (y_{i,1}, \ldots, y_{i,|y_i|})$, receive a terminal reward $r_i \in \{0, 1\}$, and define the within-group mean $\bar{r} = \frac{1}{G} \sum_{j=1}^{G} r_j$. Broadcasting the sequence-level advantage $(r_i - \bar{r})$ to all tokens (and ignoring

clipping for analysis), the per-group policy gradient and KL regularizer are

$$\nabla_\theta \mathcal{J}_{\text{GRPO}}(\theta) = \frac{1}{G} \sum_{i=1}^{G} (r_i - \bar{r}) \, v_i - \beta \, \nabla_\theta D_{\text{KL}}\big(\pi_\theta \,\|\, \pi_{\text{ref}}\big),$$

$$v_i := \frac{1}{|y_i|} \sum_{t=1}^{|y_i|} \nabla_\theta \log \pi_\theta(y_{i,t} \mid x, y_{i,<t}; \mathcal{C}).$$

(E.1)

All expectations below are over the rollout sampling procedure (and, when stated, over data $x$).

## E.2 A VARIANCE IDENTITY FOR GROUP-RELATIVE BERNOULLI REWARDS

**Lemma E.1** (Within-group Bernoulli variance). *If $r_1, \dots, r_G \overset{\text{i.i.d.}}{\sim} \text{Bernoulli}(p)$ and $\bar{r} = \frac{1}{G} \sum_{j=1}^{G} r_j$, then*

$$\mathbb{E}\big[(r_i - \bar{r})^2\big] = \text{Var}(r_i - \bar{r}) = p(1-p)\left(1 - \frac{1}{G}\right).$$

(E.2)

*Proof.* $\text{Var}(r_i) = p(1-p)$, $\text{Var}(\bar{r}) = p(1-p)/G$, and $\text{Cov}(r_i, \bar{r}) = \frac{1}{G}\text{Var}(r_i) = p(1-p)/G$. Thus $\text{Var}(r_i - \bar{r}) = \text{Var}(r_i) + \text{Var}(\bar{r}) - 2\,\text{Cov}(r_i, \bar{r}) = p(1-p)\big(1 - \frac{1}{G}\big)$. □

## E.3 A BOUND ON THE POLICY-GRADIENT MAGNITUDE

**Proposition 1** (Policy term is controlled by $p(1-p)$). *Let $g_{\text{pol}} := \frac{1}{G} \sum_{i=1}^{G} (r_i - \bar{r}) \, v_i$ be the policy term in equation E.1. Then*

$$\mathbb{E}\big[\|g_{\text{pol}}\|^2\big] \;\leq\; p(1-p)\left(1 - \frac{1}{G}\right) \, \mathbb{E}\left[\frac{1}{G} \sum_{i=1}^{G} \|v_i\|^2\right].$$

(E.3)

*Proof.* By Cauchy–Schwarz for mixed scalar/vector sums,

$$\Big\|\sum_{i=1}^{G} a_i v_i\Big\|^2 \leq \Big(\sum_{i=1}^{G} a_i^2\Big)\Big(\sum_{i=1}^{G} \|v_i\|^2\Big).$$

With $a_i = r_i - \bar{r}$ and the prefactor $1/G$, we get $\|g_{\text{pol}}\|^2 \leq \frac{1}{G^2}\Big(\sum_i (r_i - \bar{r})^2\Big)\Big(\sum_i \|v_i\|^2\Big)$. Taking expectation and using Lemma E.1, $\mathbb{E}\big[\sum_i (r_i - \bar{r})^2\big] = G\,p(1-p)\big(1 - \frac{1}{G}\big)$, yielding equation E.3. □

**Corollary E.1** (Vanishing signal at the extremes). *The upper bound in equation E.3 vanishes as $p \to 0$ or $p \to 1$, and is maximized at $p = \frac{1}{2}$.*

## E.4 REBUTTAL OF "GIVEN ENOUGH STEPS, GRPO WILL SUCCEED ANYWAY"

In a heterogeneous batch dominated by too-easy or too-hard items, $p(1-p) \approx 0$ for most samples, so equation E.3 implies

$$\mathbb{E}\big[\|g_{\text{pol}}\|^2\big] \;\approx\; 0.$$

(E.4)

Consequently, the update in equation E.1 is governed by the regularizer $-\beta \, \nabla_\theta D_{\text{KL}}(\pi_\theta \| \pi_{\text{ref}})$, which *contracts* $\pi_\theta$ back toward $\pi_{\text{ref}}$. No amount of additional steps recovers signal from items with identically zero Bernoulli variance.

## E.5 DESCENT LEMMA VIEW: EXPECTED IMPROVEMENT PER STEP

Assume $\mathcal{J}_{\text{GRPO}}$ is $L$-smooth and consider a stochastic gradient step $\theta^+ = \theta + \eta \, \widehat{g}$, where $\widehat{g}$ is an unbiased estimator of $\nabla_\theta \mathcal{J}_{\text{GRPO}}$. The descent lemma yields

$$\mathbb{E}\big[\mathcal{J}_{\text{GRPO}}(\theta^+) - \mathcal{J}_{\text{GRPO}}(\theta)\big] \;\geq\; \eta \, \big\|\mathbb{E}[\widehat{g}]\big\|^2 - \frac{L\eta^2}{2} \mathbb{E}\big[\|\widehat{g}\|^2\big].$$

(E.5)

Potential-guided sampling (defined below) increases $\big\|\mathbb{E}[\widehat{g}_{\text{pol}}]\big\|$ and decreases the fraction of near-zero-variance items, thereby increasing the first term in equation E.5 at fixed batch size. In contrast, vanilla GRPO on mixed difficulty often drives $\big\|\mathbb{E}[\widehat{g}_{\text{pol}}]\big\| \downarrow$ toward zero by equation E.3, making the KL term dominate the dynamics.

### E.6 POTENTIAL AS A PROXY FOR LEARNING SIGNAL

For each training item $x_i$, let $p_i(\theta) = \mathrm{Pr}_\theta\{r_i = 1\}$ be the success probability under the (tool-augmented) policy. Estimate it by sampling $N$ answers from a *mixture of agent frameworks* and define

$$\hat{p}_i := \frac{1}{N}\sum_{j=1}^{N} \mathbf{1}\{\text{answer } j \text{ is correct}\}, \qquad \Pi_i := 4\,\hat{p}_i\,(1-\hat{p}_i) \in [0,1]. \tag{E.6}$$

**Lemma E.2** (Concentration of the potential estimator). *For any $\epsilon > 0$, $\mathrm{Pr}\big(|\hat{p}_i - p_i| \geq \epsilon\big) \leq 2\exp(-2N\epsilon^2)$. Hence $\Pi_i$ concentrates around $4p_i(1-p_i)$ as $N$ grows.*

**Proposition 2** (Potential aligns with gradient strength). *Let $v_i$ be defined as in equation E.1. Up to the slowly-varying factor $\mathbb{E}[\|v_i\|^2]$, the bound in equation E.3 shows that $\mathbb{E}\big[\|g_{\mathrm{pol}}\|^2\big]$ is maximized when $p_i(1-p_i)$ is maximized, i.e. near $p_i = \frac{1}{2}$. Therefore $\Pi_i$ serves as a* proxy *for per-item learning signal.*

### E.7 WHY MULTI-STAGE CURRICULUM HELPS

Let $\mathcal{B}$ denote the distribution over batch items. Define $\psi(\mathcal{B}) := \mathbb{E}_{x\sim\mathcal{B}}[\,p_x(1-p_x)\,]$. From equation E.3, larger $\psi(\mathcal{B})$ increases the policy term's expected squared norm. A *potential-ranked curriculum* selects items with large $\Pi_i$ in early stages and gradually lowers the threshold, so that previously too-hard items (with small $p_i$) enter the curriculum when training has moved them into the high-variance region. Thus the curriculum maintains batches with large $\psi(\mathcal{B})$ throughout training, improving the expected gain equation E.5.

**Remark E.1** (Clipping and KL). *Our analysis ignores clipping in equation E.1; including it only reduces the policy term's magnitude, strengthening the conclusion that mixed-difficulty batches yield weak signal. The KL regularizer is essential for stability but dominates updates exactly when the policy signal is small.*

### E.8 CONCLUSIONS

- **Vanishing-signal regime.** On too-easy or too-hard items, Bernoulli variance $p(1-p)$ collapses and the policy gradient is provably tiny (Prop. 1).
- **Why vanilla GRPO underperforms.** Mixed batches over-weight near-zero-variance items; the KL term then governs the update and contracts toward $\pi_{\mathrm{ref}}$.
- **Why potential-guided curriculum works.** Selecting by $\Pi_i = 4\hat{p}_i(1-\hat{p}_i)$ targets the high-variance region where learning signal is largest (Prop. 2) and maintains it across stages, improving the expected per-step gain (Eq. equation E.5).

## F PERFORMANCE OF R1-CODE-INTERPRETER IN OUT-OF-DISTRIBUTION (OOD) TASKS

Table 4: Performance of R1-CI-14B and R1-CI-7B in OOD tasks: Graduate-Level Google-Proof Q&A (GPQA, Diamond) (Rein et al., 2024), and American Invitational Mathematics Examination (AIME 24&25).

| | Each slot is the average of three repeated runs | | | | | |
|---|---|---|---|---|---|---|
| **Model** | **Qwen-2.5-7B All Text** | **Qwen-2.5-7B CodeSteer** | **R1-CI-7B** | **Qwen-2.5-14B All Text** | **Qwen-2.5-14B CodeSteer** | **R1-CI-14B** |
| GPQA | 31.2 | 32.9 | **39.0** | 40.1 | 41.2 | **50.2** |
| AIME 2024 & 2025 | 8.33 | 8.33 | **15.0** | 30.0 | 33.3 | **42.0** |

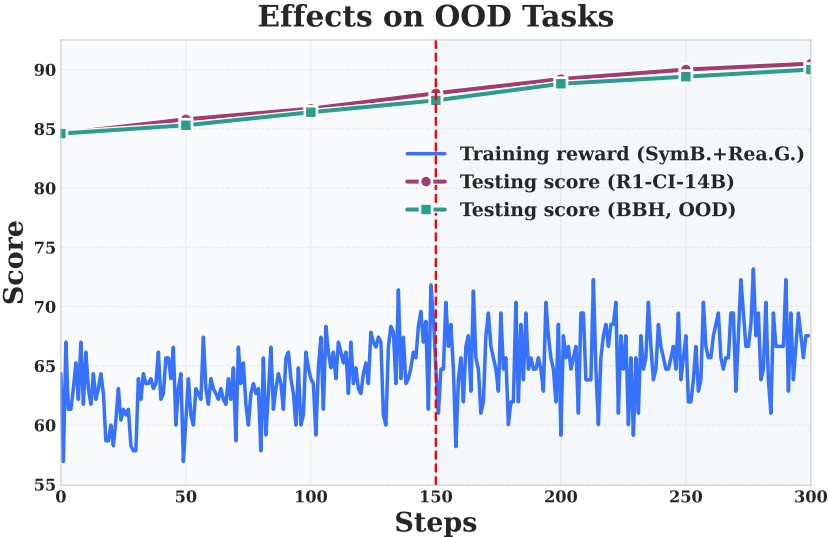

Figure 13: Evaluation on out-of-distribution (OOD) tasks. We trained the 14B model using two-stage GRPO with SymBench and Reasoning-Gym as training data, and evaluated it on Big-Bench-Hard(BBH) as an OOD benchmark. The number of training tasks remained 107. The blue curve shows the steadily increasing reward during two-stage GRPO training. Comparing the BBH test results of the new model (green curve) with the original R1-CI-14B model (purple curve) shows similar performance, indicating that our training framework generalizes effectively to unseen task distributions from varied sources.

## G   IMPACT OF SAMPLE IMPROVEMENT POTENTIAL ON GRPO TRAINING

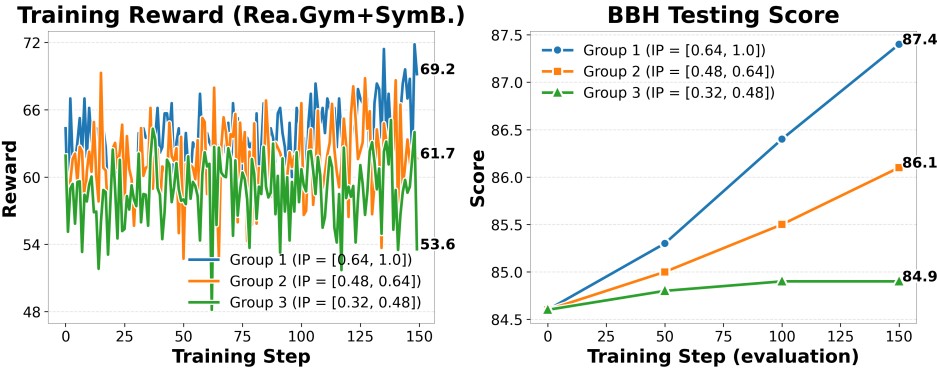

Figure 14: GRPO training using datasets grouped by improvement potential: Group 1 [0.64–1.00], Group 2 [0.48–0.64], and Group 3 [0.32–0.48]. Each group contains the same number of samples drawn from the same Reasoning Gym and SymBench tasks, but with different improvement potential ranges. Models were evaluated on BBH for fair comparison. Models trained on higher-potential samples show consistently rising training rewards and achieve higher BBH test scores.

## H   ABLATION STUDIES ON WARM START VS. COLD START, QWEN VS. DEEPSEEK-DISTILLED REASONING MODELS AS BASE MODELS

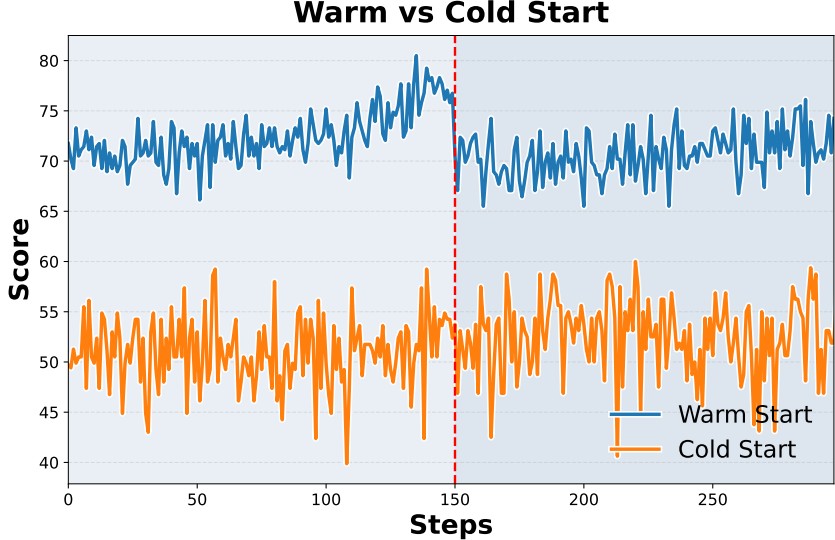

Figure 15: Warm- vs. cold-start. With GRPO, a warm start (preceded by SFT) outperforms a cold start for Qwen-14B model as the base model. The model without prior SFT process gets barely performance lifting during training, even though the training data also has been calibrated by improvement potential for multi-stage training.

Table 5: Ablation studies on using DeepSeek-distilled reasoning models as the base model.

| | Tested on 37 Test Tasks | | | | |
|---|---|---|---|---|---|
| Model | 7B, SFT | 7B, All Text | 7B, All Code | 14B, All Text | 14B, All Code |
| DeepSeek | 53.1 | 27.9 | 28.7 | 40.1 | 43.4 |
| Qwen-2.5 | **57.0** | **32.2** | **41.5** | **44.1** | **47.0** |

## I   DESCRIPTION OF REASONING AND PLANNING TASKS

Here we describe the 144 training and testing tasks. They require strong symbolic, mathematical, logical, geometrical, scientific, and commonsense reasoning capabilities. The T1 to T33 tasks originate from SymBench (Chen et al., 2025), T34 to T60 tasks originate from Big-Bench-Hard (Suzgun et al., 2022), and the last T61 to T144 tasks are from Reasoning-Gym[2]. We select questions with diverse difficulty and standardize them as a unified format to support fast rollout and testing.

**T1 - 2048**   Similarly to the 2048 game, in a grid, numbers representing powers of 2 can move in any direction, combining when they encounter a matching number to form the next power of 2. Given a starting position and a sequence of movements, the goal is to determine the resulting grid after executing the moves.

**T2 - Blocksworld**   In Blocksworld, the objective is to stack a set of blocks (brown) according to a specific order. The robot can perform four actions: (1) pick up a block, (2) unstack a block from the top of another block, (3) put down a block, (4) stack a block on top of another block. A robot can only pick up, unstack, or stack a block if it is clear, that is, the block has no other blocks on top and is not currently being held.

**T3 - BoxLift**   This task involves coordinating robots of various types to lift boxes of different sizes and weights. Each robot has a specific lifting capacity and can collaborate with others to lift a single

---

[2]https://github.com/open-thought/reasoning-gym

box. A box can only be lifted if the combined lifting capacity of the robots exceeds the box's weight. The objective is to lift all the boxes in the minimum number of time steps.

**T4 - BoxNet**    This task involves coordinating robot arms to move colored boxes (squares) into corresponding colored goal locations (circles) in the fewest time steps. Each robot arm is assigned and restricted to a cell indicated by the dotted lines. The arms have two possible actions: (1) move a box within their cell to a neighboring cell, or (2) move a box within their cell to a goal location within the same cell. The objective is to ensure all boxes are placed in their matching goal locations efficiently.

**T5 - Combinatoral Calculation**    Given a set of integers, the goal is to use arithmetic operations (addition, subtraction, multiplication, division) and parentheses to arrange the numbers in such a way that the final result matches a specified target value. Each number must be used exactly once, and the order of the numbers cannot be changed.

**T6 - Constrained Linear Arrangement**    In a two-player card game, the task is to deduce your opponent's moves based on the game's rules, your played cards, and the announced results of each round. Each card can only be used once, and the game follows specific interaction rules between different card types, where certain cards can defeat, be defeated by, or draw with others according to predefined relationships.

**T7 - Cryptanalysis**    In this task, you are provided with a combination lock consisting of numbers and letters, where neither the numbers nor the letters repeat. Using a series of guesses and feedback, the goal is to deduce the correct password based on the given conditions.

**T8 - Eight Queens**    Given a grid with some queens already placed, the task is to place the remaining queens such that no two queens share the same row, column, or diagonal, while avoiding positions with obstacles in the grid.

**T9 - Game 24**    This task involves querying LLMs to use a given set of integers to generate an equation that evaluates to 24.

**T10 - Gridworld**    This task involves querying LLMs to plan the robot actions in a grid world, reaching all goals in any order while avoiding obstacles.

**T11 - GSM** (Gao et al., 2023)    This is the more challenging version of GSM8K (Cobbe et al., 2021) math reasoning dataset, where the numbers in the original questions of GSM8K are replaced with larger, less common values.

**T12 - Letter Logic Diagram**    The task is to complete an incomplete grid by selecting from a list of letters, where each row and column must contain each letter exactly once, and all cells on the minor diagonal (top-right to bottom-left) must contain the same letter. Some cells are already filled in as constraints.

**T13 - Letters**    This task involves querying LLMs to count the total number of specific letters in a long word and specify their positions. An example question can be 'How many r's in the word strawberry and what are their positions?'. This task has recently gained significant attention because current LLMs struggle to perform it effectively and accurately.

**T14 - Light Puzzles**    In this task, you are given an $n \times n$ grid representing a network of lights, where a lit light is represented by "1" and an unlit light by "0". Several buttons control the state of these lights by turning them on or off in certain positions. The state of each light can be affected by multiple buttons. The task is to follow a series of button presses and determine the final state of the grid.

**T15 - Logical Puzzle**    The task involves querying LLMs to select a specified number of different values from a grid of numbers, ensuring that certain mathematical constraints (sum or product) are satisfied for selected numbers for each row and column.

**T16 - Logical Equation**    The task is to assign a specific numeric value to each letter from a given set, using a predefined range of numbers and a set of inequalities. Each letter corresponds to a unique number, and the relationships between the letters are defined by mathematical equations or constraints.

**T17 - Mahjong**   Given an initial set of letter cards, in each round, a new card is added and one card is removed. Some effects may happen when specific combinations of the cards appear after introducing the new card. A result is determined based on these specific conditions. The goal is to determine a result based on a series of rounds.

**T18 - MATH-Count&Probability**   This is the math reasoning dataset from MATH dataset (Hendrycks et al., 2021), with specific focus on counting and probability questions.

**T19 - MATH-Geometry**   This is the math reasoning dataset from MATH dataset (Hendrycks et al., 2021), with specific focus on geometry questions.

**T20 - Matrix Transformation**   Rotate a given matrix of characters based on given instruction (e.g., 90 degrees clockwise), preserving each character's position relative to others in the transformed output. The input matrix can be of any size and contain any character.

**T21 - New Operator**   This task introduces custom mathematical operations involving two numbers, defined with unique formulas. The goal is to use the given definitions of these operations to compute the result of a specific expression.

**T22 - Number Multiplying**   This task involves querying LLMs to compute the product among integers. It represents a classic problem that LLMs are not able to solve through pure textual reasoning.

**T23 - Pattern Recognition**   The task involves querying LLMs to find all squares in a character matrix where each square consists of identical characters and has a side length of at least 3.

**T24 - Permutation and Combination**   Given a set of objects with specific positioning constraints, the task is to determine the correct arrangement of the objects on a shelf. Each object must be placed in a position according to the rules provided, ensuring that the conditions on adjacency, order, and specific positions are met. For example, a rule about adjacency could be 'Book A must be adjacent to book I'.

**T25 - Pooling**   This task involves applying a pooling operation on a numerical $N \times N$ grid. The pooling operation uses an $n \times n$ sliding window ($n < N$) that moves across the grid from left to right and top to bottom. The results from each window are then arranged based on their positions to create a new output matrix.

**T26 - Reversi**   In this game similar to Reversi, players take turns placing pieces on an $n \times n$ grid. After placing a piece, any of the opponent's pieces located between two of the player's pieces (in the same row, column, or diagonal) will be flipped. The task is to determine the state of the board after rounds, starting from a given configuration.

**T27 - Standard Sudoku**   Given a partially filled Sudoku grid, the task is to fill the remaining empty cells with numbers between 1 and 9, ensuring that no number repeats in the same row, column, or $3 \times 3$ subgrid.

**T28 - Statistical Counting**   Calculate the total score of a string by scanning it from left to right, where consecutive identical letters earn points (for example, two or more consecutive A's add 1 point, B's add 2 points, etc.). The task is to start with a score of 0 and return the final summing value.

**T29 - String Deletion and Modification**   The task is to transform a string by repeatedly applying a set of ordered string manipulation rules until no more changes are possible, where each rule modifies the string based on specific patterns or conditions present in the current string state. For example, a modification rule can be "If the string ends with 'ba', replace it with 'ab'."

**T30 - String Insertion**   The task is to transform a string by scanning it from left to right and inserting specific characters after certain character patterns (e.g., each pattern WXYZ requires inserting W immediately after it occurs). All operations are performed simultaneously on the original string.

**T31 - String Splitting**   A dismantling engineer has old machines and can obtain machine parts through a set of predefined methods. By continuously cycling through these methods in a specific order, the engineer dismantles machines or combines parts to create new components, and the task is to determine the total number of parts and remaining machines after all possible cycles.

**T32 - String Synthesis** Given an initial set of blocks and a set of synthesis rules that combine different types of blocks, the task is to determine the final block(s) after repeatedly applying these rules in order until no more combinations are possible.

**T33 - Synthesis Decomposition** A farmer grows various crops and can exchange them for agricultural products. Using a set of methods, he can trade specific combinations of crops for products, following a cyclic pattern until no further exchanges are possible. The goal is to determine the synthesis result for each round.

**T34 - Boolean Expressions** This task determines whether a randomly generated Boolean expression—built from the constants `True` and `False` and the operators `and`, `or`, and `not`—evaluates to true or false.

**T35 - Causal Judgment** This task involves querying LLMs to read a brief scenario and predict the answer an average person would give to a causal question about it, including moral, intentional, or counterfactual aspects.

**T36 - Date Understanding** This task involves querying LLMs to interpret a few sentences that reference dates and answer a related question (e.g., compute and return a specific date in `MM/DD/YYYY` format).

**T37 - Disambiguation QA** For a sentence containing a potentially ambiguous pronoun, the task is to decide whether its reference is genuinely unclear; if it is clear, identify the noun to which the pronoun refers.

**T38 - Dyck Languages** The task aims to complete a Dyck-4 string by providing the missing closing parentheses that properly balance the given prefix.

**T39 - Formal Fallacies** The task examines a set of statements and judges whether the informal argument that follows is deductively valid or commits a formal fallacy, with particular attention to negations.

**T40 - Geometric Shapes** The task aim to analyze a full SVG `path` description and identify the geometric figure that would be drawn.

**T41 - Hyperbaton** Given two sentences, choose which of two sentences follows the natural English ordering of adjectives.

**T42-T44 - Logical Deduction 3/5/7 Objects** Use spatial clues to determine the correct ordering of a set of objects (3/5/7 objects).

**T45 - Movie Recommendation** From four candidate films, select the one that best matches the preferences implied by the movies a user has already enjoyed.

**T46 - Multi-Step Arithmetic** Perform multi-step calculations involving addition, subtraction, multiplication, and division to obtain the correct result.

**T47 - Navigate** Follow a sequence of movement instructions and state whether the agent finishes at its starting point.

**T48 - Object Counting** Given a list of items and their quantities, count how many belong to a specified category.

**T49 - Penguins in a Table** Refer to a table of individual penguins and their attributes (possibly with extra information) to answer a question about one of those attributes.

**T50 - Reasoning about Colored Objects** Using the provided context, state the color of a particular object on a surface.

**T51 - Ruin Names** Make a single-character change to an artist, band, or movie name to create a humorous new meaning.

**T52 - Salient Translation Error Detection** Examine a German sentence and its English translation, and classify the main translation error present.

**T53 - Snarks** From a pair of nearly identical sentences, identify the one that is sarcastic.

**T54 - Sports Understanding** Judge whether a fabricated sports-related statement is plausible.

**T55 - Temporal Sequences** Given a timeline of a person's daily activities, identify a time slot when they could have performed another specified task.

**T56-T58 - Tracking Shuffled 3/5/7 Objects** Trace a set of objects (3/5/7 objects) through a series of pairwise swaps to determine their final positions.

**T59 - Web of Lies** Decide whether a Boolean function described in a word problem evaluates to true or false.

**T60 - Word Sorting** Arrange the provided words in standard alphabetical order.

**T61 - AB** Rewrite an A::B token string by exhaustively applying neighbor collision rules and return the final sequence.

**T62 - Acre** From example Blicket detector outcomes, decide whether a new object set turns the detector "on", "off", or is "undetermined".

**T63 - Advanced Geometry** Solve analytic geometry questions (e.g. angle, orthocentre, in-circle radius) given vertex coordinates.

**T64 - AIW** Answer small "Alice-in-Wonderland" social reasoning problems about siblings, friends, or colleagues.

**T65 - ARC_1D** Infer the mapping rule that maps example 1D grids to output grids and apply it to a test grid.

**T66 - ARC_AGI** Same as `ARC_1D` but with rotations, mirrors and permutations on 2-D grids.

**T67 - Base Conversion** The task is to convert integers between arbitrary bases.

**T68 - Basic Arithmetic** The task is to evaluate the value of basic arithmetic expressions.

**T69 - BF** Based on outputs of example BF codes, the task is to output the contents of a new BF program.

**T70 - Binary Alternation** The task is to produce the minimum number of character swaps to make a binary string to be alternating, that is, no two adjacent characters are equal.

**T71 - Binary Matrix** Given binary matrices, the task is to find the distance to the nearest 0 for each cell in the matrix.

**T72 - Bitwise Arithmetic** The task is to compute results of expressions with mixed bitwise and arithmetic operators.

**T73 - Caesar Cipher** The task is to decrypt a Caesar cipher text.

**T74 - Calendar Arithmetic** Given a description of the calendar, answer a question by conducting arithmetic calculations such as adding or subtracting days / months / years or computing weekday differences.

**T75 - Chain Sum** The task is to calculate simple arithmetic problem and output the answer.

**T76 - Circuit Logic** Given a logic circuit with logical operators, the goal is to evaluate the output of given inputs.

**T77 - Codeio** The task is to read and reason about task description and pseudocode programs and report outputs of the program given inputs.

**T78 - Color Cube Rotation** After rotating a 3D colored cube, the task is to name the color on a queried face.

**T79 - Complex Arithmetic** The task is to perform arithmetic with complex numbers and report answers.

**T80 - Count Bits** Given a large number, the goal is to count the number of occurrence of 1 bits in the binary representation of this number.

**T81 - Count Primes** The task is to count number of primes numbers within an interval.

**T82 - Countdown** The task is to write an expression that can reach a target integer using given numbers and the four operations.

**T83 - Course Schedule** Given a list of courses need to be taken and their prerequisites, the task is to determine if all courses can be finished.

**T84 - Decimal Arithmetic** The task is to evaluate decimal expressions with given precision.

**T85 - Dice** The task is to compute probabilities of rolling results in fair-dice experiments with various dices with different number of sides.

**T86 - Emoji Mystery** The task is to deduce hidden sentences expressed with emoji symbols.

**T87 - Family Relationships** The task is to answer kinship queries in family trees.

**T88 - Figlet Font** The task is to read FIGlet banners and output the content as strings.

**T89 - Fraction Simplification** The task is to simplify fractions to the lowest terms.

**T90 - Futoshiki** The task is to fill in values to empty spaces of Futoshiki puzzles that have inequalities.

**T91 - Game of Life** The task is to simulate Conway's Game-of-Life for k steps.

**T92 - Game of Life Halting** The task is to decide whether a Game-of-Life configuration halts within k steps, that is, there are no cells alive.

**T93 - GCD** The task is to compute greatest common divisors of numbers.

**T94 - Graph Color** The task is to provide a coloring for this graph such that every vertex is not connected to a vertex of the same color.

**T95 - Group Anagrams** Given a list of words, the task is to cluster words that are anagrams.

**T96 - Intermediate Integration** Given an expression, the task is to calculate the indefinite integral.

**T97 - Isomorphic Strings** The task is to decide if two strings can be isomorphic if the characters in one string can be replaced to get the second string.

**T98 - Jugs** Given empty jugs with different sizes, the task is to give a plan of how to fill any of the available jugs with the target amount of water by filling, emptying, or pouring from a jug to another.

**T99 - Knight Swap** The task is to swap two knights on a chessboard in the fewest moves.

**T100 - Knights Knaves** The task is to determine who is a knight (truth-teller) or knave from their statements.

**T101 - Largest Island** The task is to find max connected component size in a binary grid.

**T102 - LCM** The task is to find the Least Common Multiple (LCM) of numbers.

**T103 - Leg Counting** The task is to count how many legs there are in total when given a list of animals.

**T104 - Letter Jumble** For each word in a sentence, the letter may have been randomly shuffled. The task is to reconstruct original words from jumbled letters.

**T105 - List Functions** Given examples of how inputs are mapped to outputs, reason and use the same mapping to generate output given an input.

**T106 - Manipulate Matrix** Apply a sequence of matrix transformations to a matrix and output the result.

**T107 - Maze** Compute the shortest path length from start to goal in an maze.

**T108 - Modulo Grid** Identify the mathematical pattern which defines a grid, then use that pattern to fill in the question marks in this grid.

**T109 - Needle Haystack** The task is to locate a short pattern inside a longer string.

**T110 - Number Filtering** Given a list of numbers and a requirement, remove numbers not satisfying this requirement.

**T111 - Number Format** The task is to pick the largest/smallest number out of several options.

**T112 - Number Sequence** Predict the next term of integer sequences based on previous patterns.

**T113 - Number Sorting** The task is to sort number lists based on required order.

**T114 - Palindrome Generation** The task is, given a list of letters, to form a valid palindrome.

**T115 - Palindrome Partitioning** Given a string, the task is to find all ways to partition it such that every substring is a palindrome.

**T116 - Polynomial Equations** The task is to find real values of a variable in a polynomial equation.

**T117 - Polynomial Multiplication** The task is to calculate the result of multiplying two polynomials.

**T118 - Pool Matrix** The task is to perform max- or average-pooling on numeric matrices.

**T119 - Products** The task is to compute multiplications of numbers.

**T120 - Propositional Logic** Given a list of premises, the task is to infer a correct conclusion from the premise.

**T121 - Quantum Lock** There are some buttons, a light, and a number. The light will toggle between red and green whenever you press a button. Each button performs a mathematical operation to the number, but the operation may depend on the state of the light. You must press the shortest correct sequence of buttons to reach the target value.

**T122 - Ransom Note** Given two strings representing a ransom note and a magazine, determine if the ransom note can be constructed using the letters in the magazine

**T123 - Rearc** Find the common rule that maps input grids to output grids and apply the rule to predict corresponding output of a test input grid.

**T124 - Rectangle Count** The task is to count how many rectangles are present in an ASCII grid.

**T125 - Rotate Matrix** Given a square matrix, the task is to rotate it and output the rotated matrix.

**T126 - Rotten Oranges** You are given an n x n grid where each cell can be empty cell, contain a fresh orange, or contain a rotten orange. Every minute, any fresh orange that is 4-directionally adjacent to a rotten orange becomes rotten. Your task is determine the minimum number of minutes that must elapse until no cell has a fresh orange.

**T127 - Rubiks Cube** Given a Rubik's cube, the task is to provide a solution to solve this cube using Singmaster notation.

**T128 - Rush Hour** Given a rush hour parking grid, the task is to give a plan of movements of cars to achieve required car positions.

**T129 - Self Reference** The task is to evaluate self-referential arithmetic expressions to output the number of possible solutions.

**T130 - Shortest Path** The task is to find the length of the shortest path in a grid.

**T131 - Simple Equations** The task is to solve equations with one variable.

**T132 - Simple Geometry** Given polygon with different number of sides and all interior angles but one angle, the task is to calculate the remaining interior angle.

**T133 - Simple Integration** The task is to find solution to indefinite integral problems with one variable.

**T134 - Sokoban** The task is to find a list of actions to solve a Sokoban level.

**T135 - Spell Backward** The task is to reverse input strings.

**T136 - Spiral Matrix** Given a matrix, the task is to generate a list of elements in spiral order, starting from the top-left element.

**T137 - String Manipulation** The task is to repeatedly transform a string according to a set of rules until no further transformations can be performed, or a state is repeated.

**T138 - Syllogism** Given some statements, answer the provided question by retrieving information from the statements.

**T139 - Time Intervals** The task is to compute durations between two times / dates with various formats and complexities.

**T140 - Tower of Hanoi** Output an optimal (or specified) move list to transfer disks between pegs to solve a tower of Hanoi problem.

**T141 - Tsumego** Choose the single correct Go move to capture or save stones.

**T142 - Word Ladder** Transform one word to another by single-letter changes using dictionary words.

**T143 - Word Sequence Reversal** Given a list of words, the task is to reverse order of words.

**T144 - Zebra Puzzles** Given some statements, solve the logic puzzle by gathering information from statements and deduce the answer of the question.

Table 6: The evaluated capabilities of all 144 tasks, classified as Execution, Planning, and Reasoning tasks. The classification is based on human experts' knowledge and also the classification in original datasets if it exists.

| | Task | Math | Spatial | Logical | Order | Optimization | Search |
|---|---|---|---|---|---|---|---|
| **Execution** | 2048 | ✓ | ✓ | ✓ | ✗ | ✗ | ✗ |
| | Light Puzzles | ✗ | ✓ | ✗ | ✗ | ✗ | ✗ |
| | Mahjong | ✗ | ✗ | ✗ | ✓ | ✗ | ✗ |
| | Matrix Transform. | ✗ | ✓ | ✗ | ✗ | ✗ | ✗ |
| | New operator | ✓ | ✗ | ✗ | ✗ | ✗ | ✗ |
| | Number Multiplying | ✓ | ✗ | ✗ | ✗ | ✗ | ✗ |
| | Pattern Recognition | ✗ | ✓ | ✗ | ✗ | ✗ | ✓ |
| | Pooling | ✓ | ✓ | ✗ | ✗ | ✗ | ✗ |
| | Reversi | ✗ | ✓ | ✗ | ✗ | ✗ | ✗ |
| | Statistical Counting | ✓ | ✗ | ✗ | ✓ | ✗ | ✗ |
| | String Del. &Modi. | ✗ | ✗ | ✓ | ✓ | ✗ | ✓ |
| | String Insertion | ✗ | ✗ | ✓ | ✓ | ✗ | ✓ |
| | String Splitting | ✗ | ✗ | ✓ | ✓ | ✗ | ✓ |
| | String Synthesis | ✗ | ✗ | ✓ | ✓ | ✗ | ✓ |
| | Synthesis Decomp. | ✗ | ✗ | ✓ | ✓ | ✗ | ✓ |
| | Dyck Languages | ✗ | ✗ | ✓ | ✓ | ✗ | ✗ |
| | Multi-Step Arithmetic | ✓ | ✗ | ✓ | ✗ | ✗ | ✗ |
| | Navigate | ✗ | ✓ | ✗ | ✓ | ✗ | ✗ |
| | Object Counting | ✓ | ✗ | ✓ | ✗ | ✗ | ✗ |
| | Ruin Names | ✗ | ✗ | ✓ | ✗ | ✗ | ✓ |
| | Tracking Shuffled Obj. | ✗ | ✓ | ✓ | ✓ | ✗ | ✗ |
| | Word Sorting | ✗ | ✗ | ✗ | ✓ | ✗ | ✗ |
| | AB | ✗ | ✗ | ✓ | ✓ | ✗ | ✗ |
| | ARC_1D | ✗ | ✓ | ✗ | ✓ | ✗ | ✗ |
| | ARC_AGI | ✗ | ✓ | ✗ | ✓ | ✗ | ✗ |
| | Base Conversion | ✓ | ✗ | ✗ | ✗ | ✗ | ✗ |
| | Basic Arithmetic | ✓ | ✗ | ✗ | ✗ | ✗ | ✗ |
| | BF | ✗ | ✗ | ✓ | ✓ | ✗ | ✗ |
| | Binary Alternation | ✗ | ✗ | ✓ | ✓ | ✗ | ✗ |
| | Binary Matrix | ✓ | ✓ | ✗ | ✗ | ✗ | ✓ |
| | Bitwise Arithmetic | ✓ | ✗ | ✗ | ✗ | ✗ | ✗ |
| | Caesar Cipher | ✗ | ✗ | ✗ | ✓ | ✗ | ✗ |
| | Chain Sum | ✓ | ✗ | ✗ | ✗ | ✗ | ✗ |
| | Codeio | ✗ | ✗ | ✓ | ✓ | ✗ | ✗ |
| | Color Cube Rotation | ✗ | ✓ | ✗ | ✓ | ✗ | ✗ |
| | Complex Arithmetic | ✓ | ✗ | ✗ | ✗ | ✗ | ✗ |
| | Count Bits | ✓ | ✗ | ✗ | ✗ | ✗ | ✗ |
| | Count Primes | ✓ | ✗ | ✗ | ✗ | ✗ | ✗ |

**Table 6 (continued from previous page)**

| | Task | Math | Spatial | Logical | Order | Optimization | Search |
|---|---|---|---|---|---|---|---|
| | Decimal Arithmetic | ✓ | ✗ | ✗ | ✗ | ✗ | ✗ |
| | Dice | ✓ | ✗ | ✗ | ✗ | ✗ | ✗ |
| | Fraction Simplification | ✓ | ✗ | ✗ | ✗ | ✗ | ✗ |
| | GCD | ✓ | ✗ | ✗ | ✗ | ✗ | ✗ |
| | Group Anagrams | ✗ | ✗ | ✗ | ✓ | ✗ | ✗ |
| | Intermediate Integration | ✓ | ✗ | ✗ | ✗ | ✗ | ✗ |
| | Isomorphic Strings | ✗ | ✗ | ✓ | ✗ | ✗ | ✗ |
| | Largest Island | ✗ | ✓ | ✗ | ✗ | ✓ | ✗ |
| | LCM | ✓ | ✗ | ✗ | ✗ | ✗ | ✗ |
| | Leg Counting | ✓ | ✗ | ✗ | ✗ | ✗ | ✗ |
| | Letter Jumble | ✗ | ✗ | ✗ | ✓ | ✗ | ✗ |
| | List Functions | ✓ | ✗ | ✗ | ✓ | ✗ | ✗ |
| | Manipulate Matrix | ✓ | ✓ | ✗ | ✗ | ✗ | ✗ |
| | Number Filtering | ✓ | ✗ | ✗ | ✗ | ✗ | ✓ |
| | Number Format | ✓ | ✗ | ✗ | ✗ | ✗ | ✗ |
| | Number Sorting | ✗ | ✗ | ✗ | ✓ | ✗ | ✗ |
| | Palindrome Generation | ✗ | ✗ | ✗ | ✓ | ✗ | ✗ |
| | Palindrome Partitioning | ✗ | ✗ | ✓ | ✗ | ✗ | ✗ |
| | Poly. Equations | ✓ | ✗ | ✗ | ✗ | ✗ | ✗ |
| | Poly. Multiplication | ✓ | ✗ | ✗ | ✗ | ✗ | ✗ |
| | Pool Matrix | ✓ | ✓ | ✗ | ✗ | ✗ | ✗ |
| | Products | ✓ | ✗ | ✗ | ✗ | ✗ | ✗ |
| | Rectangle Count | ✓ | ✓ | ✗ | ✗ | ✗ | ✓ |
| | Rotate Matrix | ✗ | ✓ | ✗ | ✗ | ✗ | ✗ |
| | Rotten Oranges | ✗ | ✓ | ✗ | ✗ | ✓ | ✗ |
| | Simple Equations | ✓ | ✗ | ✗ | ✗ | ✗ | ✗ |
| | Simple Geometry | ✓ | ✓ | ✗ | ✗ | ✗ | ✗ |
| | Simple Integration | ✓ | ✗ | ✗ | ✗ | ✗ | ✗ |
| | Spell Backward | ✗ | ✗ | ✗ | ✓ | ✗ | ✗ |
| | Spiral Matrix | ✗ | ✓ | ✗ | ✗ | ✗ | ✗ |
| | String Manipulation | ✗ | ✗ | ✓ | ✓ | ✗ | ✗ |
| | Time Intervals | ✓ | ✗ | ✓ | ✗ | ✗ | ✗ |
| | Word Seq. Reversal | ✗ | ✗ | ✗ | ✓ | ✗ | ✗ |
| **Planning** | Blocksworld | ✗ | ✓ | ✓ | ✗ | ✓ | ✗ |
| | BoxLift | ✗ | ✗ | ✓ | ✗ | ✓ | ✗ |
| | BoxNet | ✗ | ✗ | ✓ | ✗ | ✓ | ✗ |
| | Combinatorial Calc. | ✓ | ✗ | ✗ | ✗ | ✓ | ✗ |
| | Const. Linear Arrange. | ✗ | ✗ | ✓ | ✗ | ✗ | ✗ |
| | Cryptanalysis | ✗ | ✗ | ✓ | ✗ | ✗ | ✗ |
| | Eight Queens | ✗ | ✓ | ✗ | ✗ | ✗ | ✗ |
| | Game 24 | ✓ | ✗ | ✗ | ✓ | ✓ | ✗ |
| | Gridworld | ✗ | ✓ | ✗ | ✓ | ✗ | ✓ |
| | Letter Logic Diagram | ✗ | ✓ | ✓ | ✗ | ✗ | ✗ |
| | Letters | ✗ | ✓ | ✗ | ✗ | ✗ | ✓ |
| | Logic Puzzle | ✓ | ✓ | ✗ | ✗ | ✗ | ✓ |
| | Permut. and Combina. | ✗ | ✓ | ✓ | ✓ | ✗ | ✗ |
| | Standard Sudoku | ✓ | ✓ | ✗ | ✗ | ✗ | ✓ |
| | Movie Recommendation | ✗ | ✗ | ✗ | ✗ | ✓ | ✓ |
| | Temporal Sequences | ✓ | ✗ | ✓ | ✓ | ✗ | ✗ |
| | Countdown | ✓ | ✗ | ✗ | ✓ | ✗ | ✗ |
| | Course Schedule | ✗ | ✗ | ✓ | ✓ | ✗ | ✗ |
| | Futoshiki | ✓ | ✓ | ✓ | ✗ | ✗ | ✗ |
| | Graph Color | ✗ | ✓ | ✓ | ✗ | ✓ | ✗ |
| | Jugs | ✓ | ✗ | ✓ | ✓ | ✓ | ✗ |
| | Knight Swap | ✗ | ✓ | ✓ | ✗ | ✓ | ✗ |
| | Maze | ✗ | ✓ | ✓ | ✗ | ✓ | ✗ |

*Continued on next page*

**Table 6 (continued from previous page)**

| | Task | Math | Spatial | Logical | Order | Optimization | Search |
|---|---|---|---|---|---|---|---|
| | Modulo Grid | ✓ | ✓ | ✓ | ✗ | ✗ | ✗ |
| | Quantum Lock | ✗ | ✗ | ✓ | ✓ | ✓ | ✗ |
| | Rubiks Cube | ✗ | ✓ | ✓ | ✓ | ✗ | ✗ |
| | Rush Hour | ✗ | ✓ | ✓ | ✗ | ✗ | ✗ |
| | Shortest Path | ✓ | ✓ | ✓ | ✗ | ✓ | ✗ |
| | Sokoban | ✗ | ✓ | ✓ | ✓ | ✗ | ✗ |
| | Tower of Hanoi | ✓ | ✗ | ✓ | ✓ | ✓ | ✗ |
| | Tsumego | ✗ | ✓ | ✓ | ✗ | ✓ | ✗ |
| | Word Ladder | ✗ | ✗ | ✓ | ✗ | ✗ | ✓ |
| **Reasoning** | Logical Deduction | ✗ | ✗ | ✓ | ✗ | ✗ | ✗ |
| | GSM | ✓ | ✗ | ✓ | ✗ | ✗ | ✗ |
| | MATH-Count&Prob. | ✓ | ✗ | ✓ | ✗ | ✗ | ✓ |
| | MATH-Geometry | ✓ | ✓ | ✗ | ✗ | ✗ | ✗ |
| | Hyperbaton | ✗ | ✗ | ✗ | ✓ | ✗ | ✗ |
| | Logical Deduction | ✗ | ✓ | ✓ | ✓ | ✗ | ✗ |
| | Penguins in a Table | ✗ | ✗ | ✓ | ✗ | ✗ | ✗ |
| | Reasoning Colored Obj. | ✗ | ✓ | ✓ | ✗ | ✗ | ✗ |
| | Salient Trans. Err. Detect. | ✗ | ✗ | ✗ | ✗ | ✗ | ✓ |
| | Snarks | ✗ | ✗ | ✗ | ✗ | ✗ | ✓ |
| | Sports Understanding | ✗ | ✗ | ✓ | ✗ | ✗ | ✓ |
| | Web of Lies | ✗ | ✗ | ✓ | ✗ | ✗ | ✗ |
| | Acre | ✗ | ✗ | ✓ | ✓ | ✗ | ✗ |
| | Advanced Geometry | ✓ | ✓ | ✗ | ✗ | ✗ | ✗ |
| | AIW | ✓ | ✗ | ✓ | ✗ | ✗ | ✗ |
| | Calendar Arithmetic | ✓ | ✗ | ✗ | ✗ | ✗ | ✗ |
| | Circuit Logic | ✗ | ✗ | ✓ | ✗ | ✗ | ✗ |
| | Emoji Mystery | ✗ | ✗ | ✓ | ✗ | ✗ | ✗ |
| | Family Relationships | ✗ | ✗ | ✓ | ✗ | ✗ | ✗ |
| | Figlet Font | ✗ | ✓ | ✗ | ✓ | ✗ | ✗ |
| | Game of Life | ✗ | ✓ | ✓ | ✗ | ✗ | ✗ |
| | Game of Life Halting | ✗ | ✓ | ✓ | ✗ | ✗ | ✗ |
| | Knights Knaves | ✗ | ✗ | ✓ | ✗ | ✗ | ✗ |
| | Needle Haystack | ✗ | ✗ | ✗ | ✗ | ✗ | ✓ |
| | Number Sequence | ✓ | ✗ | ✓ | ✗ | ✗ | ✗ |
| | Propositional Logic | ✗ | ✗ | ✓ | ✗ | ✗ | ✗ |
| | Ransom Note | ✗ | ✗ | ✓ | ✗ | ✗ | ✓ |
| | Rearc | ✗ | ✓ | ✗ | ✗ | ✗ | ✓ |
| | Self Reference | ✗ | ✗ | ✓ | ✗ | ✗ | ✗ |
| | Syllogism | ✗ | ✗ | ✓ | ✗ | ✗ | ✗ |
| | Zebra Puzzles | ✗ | ✗ | ✓ | ✗ | ✗ | ✗ |

