# OpenReview forum: "R1-Code-Interpreter: LLMs Reason with Code via Supervised and Multi-stage Reinforcement Learning"
_ICLR.cc/2026/Conference — ICLR 2026 Poster_

### Official Review · Reviewer_HYQj · 2025-10-30

**Soundness:** 2
**Presentation:** 3
**Contribution:** 2
**Rating:** 4
**Confidence:** 5

**Summary:**

This paper introduces R1-Code-Interpreter, a novel framework for training Large Language Models (LLMs) to autonomously leverage a code interpreter across a diverse set of reasoning and planning tasks. The authors first identify a critical challenge: standard reinforcement learning (RL) methods like GRPO fail to yield significant improvements when applied to a heterogeneous set of tasks due to task heterogeneity and the scarcity of effective samples.
To address this, the core contribution of the paper is a multi-stage curriculum learning (CL) approach guided by "Improvement Potential." This method first uses an "Agent group" to estimate the empirical correctness rate $p_i$ for each training sample, then calculates its "Improvement Potential" $\Pi_i = 4 p_i(1-p_i)$. This metric is maximized when $p_i \approx 0.5$, i.e., when the model is "on the fence" about the sample. The authors theoretically justify this by linking it to the variance of the GRPO gradient, $p(1-p)$, where samples with $p_i \approx 0.5$ provide the strongest learning signal. The RL training curriculum proceeds in stages, starting with high-potential samples and gradually incorporating lower-potential ones.
Additionally, the paper presents a "Code Execution Sandbox" that decouples code execution from GPU gradient computation, significantly improving training efficiency (39% reduction in training time). Experimental results show that their R1-CI-14B model (based on Qwen-2.5-14B) achieves a 72.4% average accuracy on 37 test tasks, outperforming GPT-4o (58.6%) and GPT-4o with Code Interpreter (70.9%). The paper also reports an "emergent self-checking behavior," where the model learns to generate code to verify its previous reasoning steps.

**Strengths:**

1. Practical, replicable CI protocol: simple python code blocks, a clear final-answer marker, and tight caps on tool use (e.g., max code calls and per-call timeout) that help reproducibility.
2. Tangible engineering win: decoupling code execution into a CPU sandbox reduces RL training wall-clock and avoids GPU stalls.
3. Within-scope performance & diagnostics: decent gains on their own benchmark plus ablations and behavior analysis (e.g., code-based self-checking, typical call counts) that clarify where improvements come from.

**Weaknesses:**

1. Improvement Potential and curriculum learning: This looks novel at first glance, but in RL it is increasingly standard to focus on data where the model neither gets everything right nor everything wrong as the main RL signal. The paper largely describes this under a Bernoulli correctness assumption (Pi = 4p(1 - p)), which—at least to me—does not amount to a significant new idea or contribution beyond formalizing that intuition.
2. Limited evaluation breadth; same-source testing; augmented SFT without ablations: The paper evaluates on targeted splits from the same source suites and uses data augmentation for SFT. For example: “To generate SFT supervision, we prompt GPT-4o to produce multiple reasoning/execution trajectories per task and retain only those yielding correct answers. To enhance diversity and adaptability, we use varied prompt formats: some allow free-form reasoning such as the prompt in Table 1, while others enforce transitions between text and code.” However, there is no experiment isolating how much this augmentation/design choice contributes. As far as I know, data quality is a very important factor in LLM training, so the lack of targeted ablations weakens the evidence.
3. Missing comparisons to other open-source SOTA models at similar scale: The paper does not include head-to-head comparisons with strong contemporary open-source peers of comparable size (e.g., Qwen3-8B, etc.).

**Questions:**

1. Training uses 107 tasks and testing 37 tasks from the same three suites. Can you provide leave-one-benchmark results (train on two suites, test on the held-out suite) and/or results on entirely new, unseen tasks to demonstrate out-of-distribution generalization beyond the same-source split?
2. Your SFT is generated by GPT-4o with multiple trajectories per task, retaining only correct ones and varying prompt formats (including enforced text↔code transitions). How much does each design choice contribute? Please include ablations toggling (i) keep-only-correct vs. keep-all, (ii) prompt-format diversity on/off, and (iii) any multi-turn emphasis, and report the impact on accuracy, variance, and training stability.

---

> ### Author Response · Authors · 2025-11-22
> **Response to Reviewer HYQj (Q1-Q2)**
>
> We thank the reviewer for the inspiring questions and suggestions. We add the suggested experiments and respond to each point below. The **whole paper has been polished based on reviewers’ suggestions, with the changed texts colored blue**. We hope the reviewer will help re-evaluate our work given this additional context.
>
> ---
>
> ### **Q1. Improvement Potential and Curriculum Learning**
>
> **Reviewer’s concern:**
> In RL it is increasingly standard to focus on data where the model neither gets everything right nor everything wrong as the main RL signal. The paper largely describes this under a Bernoulli correctness assumption (Pi = 4p(1 - p)), which—at least to me—does not amount to a significant new idea or contribution beyond formalizing that intuition.
>
> **Response:**
> We acknowledge that filtering out overly easy and overly hard samples to improve RL training is a general idea. However, our contribution lies in **formalizing this intuition with theoretical grounding**, **deriving the improvement potential equation**, and **empirically demonstrating its superior effectiveness**.
>
> Beyond this, our work offers several **major contributions** beyond improvement potential and curriculum learning:
>
> - We are the **first to train a general Code Interpreter**, supported by a diverse benchmark of 144 tasks that we collected and developed.
> - We conduct **in-depth analyses** on the limitations of typical DeepSeek-style training for general code interpretation and validate the advantages of our **potential-driven multi-stage curriculum learning** through extensive experiments.
> - We further enhance training efficiency by introducing a **code execution sandbox** that improves GPU utilization.
>
> Overall, our primary goal and contribution are to **develop a general and effective LLM capable of strong reasoning with a Code Interpreter**, supported by both theoretical innovation and comprehensive empirical validation.
>
> ---
>
> ### **Q2. Augmented SFT without Ablations**
>
> **Reviewer’s concern:**
> SFT is generated by GPT-4o with multiple trajectories per task, retaining only correct ones and varying prompt formats (including enforced text↔code transitions). How much does each design choice contribute? Please include ablations toggling (i) keep-only-correct vs. keep-all, (ii) prompt-format diversity on/off, and (iii) any multi-turn emphasis, and report the impact on accuracy, variance, and training stability.
>
> **Response:**
> Thank you for the helpful suggestions. We have added the ablation experiments in the revised paper (**Figure 9b**, **Page 9, Lines 467–473**), and the results are summarized in the accompanying table.
>
> We compare several alternative SFT dataset designs:
>
> - **w/ Wrong Data:** In addition to the 6.5k correct answer trajectories, we include an equal number of incorrect answers synthesized by GPT-4o. This combination reduces performance by **9.8%** and increases variance from **1.7% to 5.1%**, indicating model instability caused by mixing correct and incorrect trajectories.
> - **w/o Varied Prompts:** The 6.5k SFT dataset is synthesized using a single generic prompt, without varying code-use prompts or strategies. Results show that prompt diversity is crucial, improving training and testing scores by **5.3%** and **4.5%**, respectively.
> - **w/o Multi-Turn Emphasis:** We remove the emphasis on multi-turn trajectories and adaptive solving strategies, keeping their ratio unchanged. The whole dataset size remains as 6.5k. This leads to performance drops of **4.4% (training)** and **3.0% (testing)**, demonstrating that diverse, high-quality multi-turn data are essential for enhancing model capability.
>
> | **Setting** | **Training Score** | **Testing Score** |
> |--------------|--------------------|--------------------|
> | Original SFT | **57.6**±1.4 | **57.0**±1.7 |
> | w/ Wrong Data | 48.5±4.3 | 47.2±5.1 |
> | w/o Varied Prompts | 52.3±1.6 | 52.5±1.4 |
> | w/o Multi-Turn Emphasis | 53.2±1.5 | 54.0±1.7 |
>
> ---

---

> > ### Comment · Reviewer_HYQj · 2025-11-24
> >
> > Thank you for the detailed response. The additional experiments indeed make the contribution of each data-processing step much clearer.
> >
> > For the theoretical contribution, I would still like to see some more direct empirical evidence for the practical significance of the analysis in Section 4.3. In particular, you conclude that “this formalizes our intuition and analysis in equation (4.5) that the greatest improvement arises from samples with balanced successes and failures.” It would be very helpful to include an experiment that explicitly compares the effect of training on samples with balanced successes and failures versus unbalanced successes and failures.
> >
> > I would also suggest using more moderate, randomly unbalanced settings (rather than an extreme case where almost all outcomes are successes), so that any performance differences can be more clearly attributed to the balance between successes and failures.

---

> ### Author Response · Authors · 2025-11-22
> **Response to Reviewer HYQj (Q3-Q4)**
>
> ### **Q3. Lack of Head-to-Head Comparisons with Contemporary Open-Source models**
>
> **Reviewer’s concern:**
> The paper does not include head-to-head comparisons with strong contemporary open-source peers of comparable size (e.g., Qwen3-8B, etc.).
>
> **Response:**
> In the revised paper (**Table 3**, **Page 7, Lines 343–346**), we added the suggested experiments comparing **Qwen3-14B** using both **single-turn All Text** and **multi-turn CodeSteer** as generation frameworks.
>
> Although Qwen3-14B (62.5, 64.4) performs significantly better than Qwen2.5-14B (47.3, 48.7), our **R1-CI-14B (68.8, 72.4)** model still achieves notably higher performance on all three benchmarks, with average training and testing scores **6.3% and 8.0% higher**, respectively.
>
> ---
>
> ### **Q4. Limited Evaluation Breadth and Same-Source Testing**
>
> **Reviewer’s concern:**
> Training uses 107 tasks and testing 37 tasks from the same three suites. Can you provide leave-one-benchmark results (train on two suites, test on the held-out suite) and/or results on entirely new, unseen tasks to demonstrate out-of-distribution (OOD) generalization beyond the same-source split?
>
> **Response:**
> Thank you for the constructive advice. We note that tasks in **SymBench**, **Reasoning-Gym**, and **Big-Bench-Hard (BBH)** are already highly distinctive, even within the same benchmark. Thus, tasks from these benchmarks can reasonably be considered OOD.
>
> To further evaluate generalizability under more challenging OOD settings, we added two additional **OOD evaluation experiments** as suggested:
>
> **1. Evaluation on New Unseen Benchmarks**
>
> In the revised paper (**Appendix Table 4**, **Page 18**, and **Discussion in Lines 395–398, Page 8**), we assess **R1-CI-7B** and **R1-CI-14B** on **Graduate-Level Google-Proof Q&A (GPQA, Diamond)** and the **American Invitational Mathematics Examination (AIME 2024 & 2025)**—two widely used reasoning benchmarks.
>
> | **Model** | **Qwen-2.5-7B All Text** | **Qwen-2.5-7B CodeSteer** | **R1-CI-7B** | **Qwen-2.5-14B All Text** | **Qwen-2.5-14B CodeSteer** | **R1-CI-14B** |
> |------------|--------------------------|----------------------------|---------------|-----------------------------|------------------------------|---------------|
> | **GPQA** | 31.2 | 32.9 | **39.0** | 40.1 | 41.2 | **50.2** |
> | **AIME 2024 & 2025** | 8.33 | 8.33 | **15.0** | 30.0 | 33.3 | **42.0** |
>
> *Each result is averaged over three runs.*
> Both models outperform their untrained counterparts, confirming strong OOD generalization.
>
> **2. Leave-One-Benchmark Evaluation**
>
> The revised **Appendix Figure 13 (Page 18)** and **Discussion (Lines 398–402, Page 8)** evaluate the generalizability of our framework by **training the 14B model on SymBench and Reasoning-Gym, then testing it on BBH** as an OOD benchmark.
>
> | **BBH Testing Score over Training Steps** | Step 0 | Step 50 | Step 100 | Step 150 | Step 200 | Step 250 | Step 300 |
> |------------------------|--------|----------|-----------|-----------|-----------|-----------|-----------|
> | R1-CI-14B | 84.6 | 85.8 | 86.7 | 88.0 | 89.2 | 90.0 | 90.5 |
> | OOD (SymBench + Reasoning-Gym) | 84.6 | 85.3 | 86.4 | 87.4 | 88.8 | 89.4 | 90.0 |
>
> The blue curve in Figure 13 shows a steady reward increase during two-stage GRPO training, while the green curve (OOD) remains comparable to the purple (original R1-CI-14B), indicating **robust generalization**.
>
> **Conclusion:**
> These results collectively demonstrate the **strong generalization ability of R1-Code-Interpreter** to unseen tasks from diverse sources.
>
> ---
>
> **We sincerely appreciate the reviewer’s constructive feedback and would be happy to discuss further with the reviewer for any additional questions or suggestions.**

---

> ### Comment · Reviewer_HYQj · 2025-11-24
>
> Thank you for the additional experiments. I think the current results do provide strong empirical support for the effectiveness of your method.
>
> I noticed in Table 3 that for the 14B setting, **R1-CI w/o GRPO** performs worse than **Qwen3-14B All Text**. Does this suggest that **GRPO** is actually the key factor driving the improvement, at least in this configuration?
>
> I also have a question about the fairness of the comparison. You are comparing a model that has been further trained on newly constructed data to a baseline that has not received any targeted data augmentation or task-aligned training. To me, this makes it difficult to isolate the specific benefit of your proposed data and training scheme. **A more appropriate baseline, in my view, would be to apply GRPO (or an equivalent training recipe) on the baseline model using your constructed data**, and then evaluate all models with the same All Text inference setting. This would better disentangle the effects of GRPO and your  design.
>
> To be clear, I am not denying your contribution of “training a general Code Interpreter across multiple tasks and domains”; I fully acknowledge this as an important aspect of your work. My concern is only about whether the current experimental setup yields a fully fair comparison. I would be happy to discuss with you whether the additional experiments I suggest are feasible and necessary, and how they might help sharpen the empirical claims of the paper.

---

> > ### Author Response · Authors · 2025-11-27
> > **Response to Reviewer HYQj - Round 2 (Q1-Q2)**
> >
> > We thank the reviewer for the prompt and constructive feedback. We have added new experiments comparing GRPO training with different improvement-potential samples to further validate our conclusions. We also provide detailed responses to the other two questions. These additions and analyses are included in the revised paper. We would be glad to discuss any remaining concerns with the reviewer.
> >
> > ---
> >
> > ### **Q1. Empirical Validation of Improvement Potential (Section 4.3)**
> >
> > **Reviewer’s concern:**
> > For the theoretical contribution, the reviewer requests **more direct empirical evidence** demonstrating the practical significance of the analysis in Section 4.3—specifically, whether samples with **balanced successes and failures** indeed yield greater improvement. The reviewer also suggests avoiding extreme distributions and instead evaluating **moderately unbalanced settings** to isolate the effect of balance.
> >
> > **Response:**
> > We appreciate the reviewer’s insightful suggestion. As recommended, we have added the corresponding comparison experiments in the revised paper (**Lines 491–495** and **Appendix Section E, page 19**). Following the reviewer’s guidance, we construct **datasets of equal size** but with **different improvement-potential ranges**, reflecting **moderately unbalanced** rather than extreme scenarios:
> >
> > - **[0.64, 1.00]**
> > - **[0.48, 0.64]**
> > - **[0.32, 0.48]**
> >
> > Training samples are drawn from **Reasoning Gym** and **SymBench**, and evaluation is conducted on **BBH** for a fair comparison across models.
> >
> > **Key findings:**
> >
> > - Models trained on **higher-potential samples ([0.64, 1.00])** exhibit **larger increases in training rewards**.
> > - These models also achieve **higher BBH scores** compared to those trained on lower-potential ranges.
> > - This provides **direct empirical support** for our theoretical analysis in Section 4.3 and confirms that **samples with more balanced successes/failures contribute more effectively to GRPO learning**.
> >
> > Overall, these results validate the effectiveness of the proposed **multi-stage improvement-potential-guided curriculum** for GRPO and strengthen the connection between our theoretical insights and practical gains.
> >
> > ---
> >
> > ### **Q2. Whether GRPO Is the Key Factor Behind the Improvement**
> >
> > **Reviewer’s concern:**
> > Table 3 shows that for the 14B setting, **R1-CI w/o GRPO performs worse than Qwen3-14B All Text**. Does this imply that **GRPO is the primary driver** of the performance gains in this configuration?
> >
> > **Response:**
> > We appreciate the reviewer’s careful observation. However, the comparison in Table 3 should be interpreted with the difference in **base model strength** in mind.
> >
> > Our GRPO training is built upon the **Qwen2.5-14B/7B/3B** series. Following the reviewer’s suggestion, we also include **Qwen-3-14B All Text** as a reference. Since **Qwen-3-14B is substantially stronger than Qwen-2.5-14B**, it is expected that:
> >
> > - **Qwen-3-14B All Text outperforms R1-CI w/o GRPO**,
> >   not because GRPO is the only effective component,
> >   but because Qwen-3 is a significantly stronger base model than Qwen-2.5.
> >
> > To provide a fair comparison, Table 3 also includes **Qwen2.5-14B All Text**. Under this matched-base setting:
> >
> > - **R1-CI w/o GRPO significantly outperforms Qwen2.5-14B All Text**,
> >   demonstrating the **substantial effectiveness of our SFT stage**.
> >
> > Furthermore:
> >
> > - **R1-CI-14B**, although based on **Qwen-2.5-14B**,
> >   **outperforms Qwen-3-14B after applying both SFT and GRPO**,
> >   highlighting the **combined power of our SFT and GRPO training pipeline**.
> >
> > Finally, because **Qwen-3 was not released** at the time we conducted this work, we could not train R1-CI on it. But we expect that:
> >
> > - **Our method would yield even stronger results when using Qwen-3 as the base model**,
> >   and the conclusions of the paper would still hold.
> >
> > ---

---

> > > ### Author Response · Authors · 2025-11-27
> > > **Response to Reviewer HYQj - Round 2 (Q3)**
> > >
> > > ### **Q3. Fairness of Comparison and Isolating the Effects of Our Training Scheme**
> > >
> > > **Reviewer’s concern:**
> > > The reviewer notes that our model is further trained on newly constructed data, whereas the baseline has not received any targeted augmentation or task-aligned training. This raises a concern about whether the comparison is fully fair. The reviewer suggests applying **GRPO (or an equivalent recipe)** to the baseline model using our constructed data, then evaluating all models under the same **All Text** inference setting to disentangle the contributions of GRPO and our design.
> > >
> > > **Response:**
> > > We appreciate the reviewer’s thoughtful concern regarding fairness. Our goal is to develop an **open-source model capable of general reasoning with a Code Interpreter**, and therefore we must train on **open-source base models** such as the Qwen series.
> > >
> > > In contrast, **GPT-4o is not open-sourced**, and neither its datasets nor its detailed training pipeline for Code Interpreter are publicly available. However, GPT-4o is a strong proprietary model **explicitly trained to use Code Interpreter**. Therefore, we compare with **GPT-4o + Code Interpreter** to evaluate whether our **training framework and constructed datasets** enable an open-source model to reach or surpass this capability.
> > >
> > > Our results show that:
> > >
> > > - **R1-CI-14B surpasses GPT-4o + Code Interpreter** on both **in-distribution** and **out-of-distribution** tasks.
> > > - This is particularly significant because **GPT-4o is much larger than 14B**.
> > >
> > > These results strongly support the **effectiveness of our method**, even under a size disadvantage.
> > >
> > > At the same time, we fully agree with the importance of isolating the effects of different components. Toward this end:
> > >
> > > - We perform comparisons **within the same model family**, i.e., Qwen2.5-14B vs. Qwen2.5-14B + SFT vs. Qwen2.5-14B + SFT + GRPO.
> > > - These intra-family comparisons allow us to **cleanly measure the contributions** of each component (SFT, GRPO, curriculum design, etc.).
> > > - As shown in our experiments, each stage provides **clear additive benefits**, demonstrating the effectiveness of the proposed design.
> > >
> > > We also apply the **same training recipe** to multiple base models (**Qwen 3B / 7B / 14B**). The consistent improvements across scales further demonstrate the **robustness and generalizability** of our pipeline.
> > >
> > > Finally, while one could attempt to directly compare differently trained models across different base architectures, such comparisons may be inherently unfair because the **underlying base capabilities differ substantially**. Therefore, we focus on both:
> > >
> > > 1. **Fair within-base comparisons** (to isolate method components), and
> > > 2. **Across-model comparisons** with GPT-4o + CI (to evaluate whether our method reaches competitive practical performance).
> > >
> > > Taken together, we believe the experimental setup is fair and effectively demonstrates the **benefits, components, and practicality** of our proposed training scheme.
> > >
> > > ---
> > >
> > > We hope our above new results and answers have addressed reviewers' concerns, and the reviewer could help re-evaluate our work. We are glad to discuss more with the reviewer. Thank you for your time and patience.

---

> > > > ### Comment · Reviewer_HYQj · 2025-11-28
> > > >
> > > > Thank you for the detailed clarification. However, my original concern about fairness is still not fully resolved. In the current “All Text” comparison, the baseline has not undergone any additional training or alignment on your constructed data, whereas R1-CI has. As a result, the comparison cannot cleanly disentangle the contribution of your proposed training scheme from the effect of simply exposing a model to the synthetic data via GRPO. From my perspective, a fairer and more informative setting would be to also apply GRPO (or a comparable recipe) to the All Text baseline on your constructed data, and then compare all models under the same All Text inference protocol.

---

> ### Author Response · Authors · 2025-11-28
> **Response to Reviewer HYQj - Round 3**
>
> Thank you for the quick response and the additional clarification. After reviewing your explanation, we now may better understand your concern. If we understand correctly, you are suggesting that, for a fair comparison validating our framework, we should fine-tune the **same base model** using the **All-Text inference protocol** and compare it directly against our **multi-turn Code Interpreter protocol**, to confirm that the framework’s setting is correct.
>
> Actually, **we have already conducted this ablation study** in the revised paper (Lines 429–431 and Fig. 9a). Specifically, we fine-tune the same base model with the same amount of 6.5k SFT samples from identical sources of datasets under three settings: **multi-turn Code Interpreter**, **single-turn All Text**, and **single-turn All Code**. **While the models perform similarly before training as shown in Fig. 9a, the multi-turn Code Interpreter setting achieves significantly better performance after training**, surpassing both All Text and All Code. This demonstrates that the multi-turn Code Interpreter protocol is indeed the superior setup.
>
> **We are not fully certain we have captured your concern correctly. If not, please feel free to clarify further so we can address it more accurately. Thank you!**

---

### Official Review · Reviewer_Eyn2 · 2025-10-31

**Soundness:** 3
**Presentation:** 4
**Contribution:** 3
**Rating:** 8
**Confidence:** 4

**Summary:**

The authors introduce R1-Code-Interpreter, training a single LLM to orchestrate natural-language reasoning with Python execution across 144 heterogeneous task families. The training pipeline couples 6.5k multi-turn SFT traces with GRPO-based RL, and its key ingredient is an improvement-potential curriculum that prioritizes items the model currently solves ~50% of the time, maximizing learning signal. Code is executed in a CPU sandbox (with an 8-call cap) to keep GPUs saturated while enabling multi-turn tool use. On 37 held-out tasks, the 14B model improves from 44.1% to 72.4%, narrowly surpassing GPT-4o + Code Interpreter. The authors conduct ablations, showing that vanilla RL plateaus on batches that are either too easy or too hard, while potential-guided scheduling sustains informative variance. The authors also characterize code-call counts, response lengths and report emergent self-checking where the model writes code to verify its own outputs. The authors provide this as evidence for a multi-stage, code interpreter-centric training paradigm in LLM reasoning research.

**Strengths:**

**Comprehensiveness:**
The authors provide a broad, carefully controlled experimental program, spanning diverse task families, staged training with curriculum learning, warm-start ablation, strong baselines, and behavioural system measurements (emergent self-checking, code-usage, verbosity), that provides compelling evidence of the paper’s general-purpose claims.

**Unlocking Code Interpreter Potential:**
This work offers a concrete, scalable recipe that elevates code execution from a math-only crutch to a general-purpose reasoning tool, yielding sizable gains across heterogeneous tasks and setting a practical foundation for the research community.

**Technical & Artifact Contribution:**
The authors commit to open-sourcing the code, model checkpoints, and datasets, and they document clearly the 144-task suite and the SFT/GRPO dataset-synthesis pipeline, materially strengthening reproducibility and the research community further studies.

**Weaknesses:**

**Single Scope Language:**
The paper aims for a general code interpreter for code generation and reasoning across tasks and domains, but trains and evaluates only with a Python executor; transfer to other languages/runtimes is untested. Identifying details on different languages would materially strengthen the general code-generation claim.

**Questions:**

Your reward seems largely binary (correct/incorrect) with small format/turn terms. Did you fine-graded or curriculum-aware reward shaping (unit-test pass fraction, etc)?

How sensitive are learning curves and final accuracy to reward design, and did you observe reward-hacking (optimizing for format/turns over task progress)?

Small Comments:
- Paper mentioned (15 agents), but I'm unsure? Did the authors mean N=20?
- Typo: polynimial equation -> polynomial

---

> ### Author Response · Authors · 2025-11-22
> **Response to Reviewer Eyn2**
>
> We thank the reviewer for the appreciation of our work. We add the suggested experiments based on suggestions from reviewers and respond to each point below. The **whole paper has been polished based on reviewers’ suggestions, with the changed texts colored blue**. We are glad to discuss more with the reviewer about this work.
>
> ---
>
> ### **Q1. Single-Scope Language**
>
> **Reviewer’s concern:**
> The paper aims for a general code interpreter for code generation and reasoning across tasks and domains, but trains and evaluates only with a Python executor; transfer to other languages/runtimes is untested. Identifying details on different languages would materially strengthen the general code-generation claim.
>
> **Response:**
> Thank you for the constructive suggestion. In our work, we focus on **Python executors**, as Python is currently the most widely used and versatile programming language. Its ability to import diverse functional packages makes it sufficiently powerful and complex for our study.
>
> For **future extensions**, we plan to integrate additional languages, for example, shell code to interface with cluster systems, and explore applying our framework to **other programming languages such as C, Java, and PyTorch**, to further broaden the capabilities of LLM-based code interpretation.
>
> ---
>
> ### **Q2. Reward Design — Binary vs. Fine-Grained Shaping**
>
> **Reviewer’s concern:**
> Your reward seems largely binary (correct/incorrect) with small format/turn terms. Did you explore fine-grained or curriculum-aware reward shaping (e.g., unit-test pass fraction, etc.)?
>
> **Response:**
> In our training setup, each task contains **50 samples of varying complexity**, each with a unique correct answer. Thus, using a **binary correctness reward** is both natural and widely adopted in reasoning model training with verifiable rewards.
>
> For many tasks, the solution involves **multi-turn code execution and textual reasoning**, making single code-based unit-test scores unsuitable for measuring partial correctness. Moreover, the **success rate across multiple samples** within the same task already provides a fractional measure of correctness, effectively serving a similar role as fine-grained unit-test–based scoring.
>
> ---
>
> ### **Q3. Sensitivity to Reward Design and Reward Hacking**
>
> **Reviewer’s concern:**
> How sensitive are learning curves and final accuracy to reward design, and did you observe reward-hacking (optimizing for format/turns over task progress)?
>
> **Response:**
> Thank you for the insightful question. In our experiments, the **training curves remain stably ascending** when the final outcome reward (+1.0) is significantly larger than the format reward (+0.1) and multi-turn penalty (−0.1). This indicates that model performance is **not highly sensitive to reward weighting**.
>
> As shown in prior studies, reward hacking can occur when models over-optimize for format rewards or avoid multi-turn reasoning to reduce penalties. However, in our setup:
>
> - The **pre-trained Qwen models** already produce well-structured text and Python code, so the format reward quickly saturates.
> - The **multi-turn penalty** (−0.1) discourages unnecessary turns without limiting reasoning depth.
> - We cap the number of turns at **6**, which prevents infinite loops or degenerate behaviors while allowing meaningful multi-step reasoning.
>
> As a result, the model **learns robustly without reward hacking behaviors** such as avoiding generation or prioritizing formatting over reasoning.
>
> ---
>
> ### **Small Comments**
>
> **Reviewer’s notes:**
> - Paper mentioned “15 agents,” but I’m unsure—did the authors mean N=20?
> - Typo: *polynimial equation* → *polynomial equation*
>
> **Response:**
> Thank you for pointing this out. We have corrected the inconsistencies and typos in the revised version:
> - “15” has been **corrected to 4**, and
> - “polynimial” has been **corrected to polynomial**.
>
> ---
>
> **We sincerely appreciate the reviewer’s constructive feedback and are happy to discuss further improvements or clarifications.**

---

> > ### Comment · Reviewer_Eyn2 · 2025-11-26
> > **Official Comment by Reviewer Eyn2**
> >
> > I thank the authors for their detailed response. My original concerns were primarily about the scope of the code interpreter being restricted to Python, the reward design and potential for reward hacking. Given the breadth of the 144-task suite, I am comfortable treating multi-language support as future work rather than a requirement for this submission.
> >
> > The qualitative arguments about format rewards saturating quickly, multi-turn penalty, and the cap on the number of turns help mitigate obvious reward-hacking modes. Further exploration of reward-weight sensitivity and degenerate behaviours would strengthen this aspect of the work, and I encourage the authors to frame this as a limitation and a potential future direction in the paper.
> >
> > Overall, my concerns have been addressed for this submission. I maintain my original positive assessment and score.

---

> > > ### Author Response · Authors · 2025-11-27
> > > **Thank you for your time for helpful review and appreciation into our work**
> > >
> > > We sincerely appreciate the time and efforts from the reviewer for constructive suggestions. We will definitely include the above discussion as limitation and future work in the final version of this paper.
> > >
> > > Best wishes,
> > > Authors of R1-Code-Interpreter

---

### Official Review · Reviewer_Rp2m · 2025-11-09

**Soundness:** 3
**Presentation:** 2
**Contribution:** 2
**Rating:** 4
**Confidence:** 4

**Summary:**

This paper introduces R1-Code-Interpreter, a framework to train text-only Large Language Models (LLMs) to effectively utilize a Code Interpreter by using multi-turn supervised fine-tuning (SFT) and reinforcement learning (RL). Addressing the challenges of task heterogeneity and a scarcity of effective samples when training across 144 diverse reasoning tasks, the authors propose a novel multi-stage curriculum learning approach. This curriculum prioritizes training samples based on their measured "improvement potential" (IP)—focusing on samples where the model's success is mixed rather than trivially easy or excessively difficult—which boosts RL gains from +3.4% to +9.3%. The final model, R1-CI-14B, significantly improves average accuracy on test tasks to 72.4%, outperforming both text-only GPT-4o (58.6%) and GPT-4o with its Code Interpreter (70.9%), while also exhibiting an emergent self-checking behavior through code generation.

**Strengths:**

1. This paper extends the task of combining symbolic code generation with text with reasoning to broader benchmarks, providing a thorough investigation of the generalizability of this paradigm.

2. This paper demonstrates innovation by proposing a curriculum learning method based on Improvement Potential, successfully extending TIR training from single-task settings to multi-task scenarios.

3. Compared to traditional curriculum learning approaches, the proposed method is designed based on improvement potential, enabling effective adaptation to TIR tasks.

4. Comprehensive evaluation across multiple benchmarks demonstrates the effectiveness of the proposed training method.

**Weaknesses:**

1. The improvement potential score defined in this paper is modeled as a function symmetric about p=½. In this case, even if two samples have the same improvement potential score, their empirical correctness rates may differ significantly. Therefore, in curriculum learning, simply incorporating samples with low improvement potential scores may overlook the training contribution differences brought by samples with different empirical correctness rates. For example, training samples with low empirical correctness rates focuses on enhancing the model's ability to handle complex problems, while training samples with high empirical correctness rates focuses on improving the model's ability to handle simple problems.

2. Curriculum learning is the core contribution of this paper, yet the paper lacks detailed ablation studies on curriculum learning rounds, only showing a single RL validation curve.

3. The paper lacks exploration of the method's adaptability to other algorithms beyond GRPO, such as Reinforce++, ARPO, CISPO, etc.
In Figure 3(b), the distinguishability between different curves is poor. I suggest trying more discriminative visualization methods (e.g., using logarithmic scale).

4. In curriculum learning, the model used to compute the improvement potential score of samples is the initial SFT model, while the model being trained is not the SFT model. Therefore, this curriculum learning method is an offline reinforcement learning approach, and I believe its training effectiveness may differ from online reinforcement learning methods.

**Questions:**

See weakness and follow questions：

1. Why is it assumed that sample rewards follow a Bernoulli distribution? Given the diversity of tasks and the complexity of model parameters, the reward distribution of samples may be complex and difficult to represent explicitly. I am uncertain whether modeling it as a Bernoulli distribution is justified.

2. What is the criterion for incorporating new lower-potential groups at each stage of curriculum learning? Specifically, at each stage, what improvement potential score threshold determines which samples are included? What is the data volume at each stage? This aspect is not clearly explained in the paper.

While the method demonstrates good generalizability and methodological soundness, and I believe it would be effective, I have concerns regarding the experimental rigor and the support for some claims. If these concerns are adequately addressed, I would consider raising my score accordingly.

---

> ### Author Response · Authors · 2025-11-22
> **Response to Reviewer Rp2m (Q1-Q4)**
>
> We thank the reviewer for the thoughtful questions and constructive suggestions. We have incorporated the requested experiments and clarifications as detailed below. The **whole paper has been polished based on reviewers’ suggestions**, with the **changed texts colored blue**. We hope these updates help the reviewer re-evaluate our work favorably.
>
> ---
>
> ### **Q1. Same improvement potential score but different empirical correctness rates**
>
> **Reviewer’s concern:**
> Even if two samples share the same improvement potential score, their empirical correctness rates may differ significantly, possibly causing unbalanced contributions during training.
>
> **Response:**
> Thank you for the insightful comment. Our **multi-stage GRPO training** merges groups progressively at each stage rather than training them separately. By the final stage, the model is trained on **all** samples (both simple and hard).
>
> - **Progressive Merging:**
>   The curriculum proceeds by adding new (lower-potential) samples at each stage and merging them with previous ones. Thus, as training advances, the dataset expands until all samples are included. This ensures learning from the full data distribution (see Page 6, Lines 316–323).
>
>   Figure 4 shows continuous improvement during stages 3 and 4, confirming that exposure to all samples enhances robustness and generalization.
>
> - **Balanced Sample Addition:**
>   Since improvement potential \( 4p(1-p) \) peaks at medium difficulty, both easy and hard samples can share similar potential scores and appear in the same group. Each stage therefore maintains a balanced mix of easy and hard samples, ensuring diverse learning dynamics.
>
> ---
>
> ### **Q2. Lack of ablation studies on RL curriculum learning rounds**
>
> **Reviewer’s concern:**
> The paper presents only one RL validation curve and lacks detailed ablation studies on RL curriculum learning rounds.
>
> **Response:**
> We appreciate the suggestion. Our curriculum training merges data at each stage, continuing from previous progress rather than restarting. Therefore, a single complete round of RL training already covers the full dataset.
>
> Following the reviewer’s advice, we conducted an additional experiment with a **second training round** (see **Figure 10 in Page 9, Lines 491–496 in Page 10**). In this experiment, we extended RL training using a newly partitioned dataset derived from the first-round model. The results show **no further improvemen**t. In fact, during the first stage, the **testing score decreases** even as the training reward increases, likely due to **overfitting** on the limited initial data group. This effect is partially mitigated in later stages as more data are added. Overall, these findings confirm that **one round of RL training is sufficient**, as the model already generalizes effectively across the full dataset.
>
> | GRPO, **second round** | Step 0 | Step 50 | Step 100 | Step 150 | Step 200 | Step 250 | Step 300 | Step 350 | Step 400 | Step 450 | Step 500 | Step 550 | Step 600 |
> |-------------------------|---------|----------|-----------|-----------|-----------|-----------|-----------|-----------|-----------|-----------|-----------|-----------|-----------|
> | **Testing score**       | 72.3 | 71.3 | 71.1 | 71.0 | 71.5 | 71.8 | 72.0 | 72.1 | 72.2 | 72.1 | 72.1 | 72.2 | 72.2 |
> ---
>
> ### **Q3. Adaptability to other RL algorithms**
>
> **Reviewer’s concern:**
> The adaptability of the proposed method to algorithms beyond GRPO (e.g., Reinforce++, ARPO, CISPO) is unclear.
>
> **Response:**
> We have added **ablation experiments comparing GRPO, PPO, and Reinforce++** (see **Figure 7**, **Page 9, Lines 479–482**). Their performances are comparable in both training reward and test score trends:
>
> | Test score | Step 0 | Step 50 | Step 100 | Step 150 |
> |------------|---------|----------|-----------|-----------|
> | GRPO | 60.6 | 61.3 | 62.5 | 65.5 |
> | PPO | 60.6 | 60.9 | 61.8 | 64.8 |
> | Reinforce++ | 60.6 | 60.9 | 63.1 | 65.0 |
>
> These results demonstrate that our **curriculum framework is compatible across different RL algorithms**.
>
> ---
>
> ### **Q4. Poor distinguishability in Figure 3(b)**
>
> **Reviewer’s concern:**
> The distinguishability between curves in Figure 3(b) is poor; consider using a logarithmic scale.
>
> **Response:**
> Thank you for the helpful suggestion. In **Figure 3(b)**, our goal is to illustrate that different tasks exhibit **diverse training trends** (some increasing, others decreasing), rather than consistent behavior. As a result, the **overall average (bold black line)** across all 144 tasks shows no clear improvement trend. Although the current figure has limited visual distinction among individual curves, it still effectively conveys this conclusion.
>
> We also tested a **logarithmic-scale plot** as suggested, but the distinguishability remained poor because the task scores **span the full 0–100 range**. Thus, the low visual clarity arises from the wide score distribution, not from curve overlap due to magnitude compression.
>
> ---

---

> ### Author Response · Authors · 2025-11-22
> **Response to Reviewer Rp2m (Q5-Q7)**
>
> ### **Q5. Offline vs. online reinforcement learning**
>
> **Reviewer’s concern:**
> In curriculum learning, the model used to compute the improvement potential score of samples is the initial SFT model, while the model being trained is not the SFT model. Therefore, this curriculum learning method is an offline reinforcement learning approach. Its effectiveness relative to online methods should be discussed.
>
> **Response:**
> Indeed, our **multi-stage RL training uses a fixed pre-partitioned dataset**, making it an offline setup. Following the reviewer’s suggestion, we added **comparisons with two online variants** (see **Figure 8 in Page 9** and **Lines 483–490 in Page 10**):
>
> 1. **Online-Merge:** Before each stage, the dataset is **repartitioned using the current model**, and training proceeds on the updated groups. As in the offline setup, later stages merge data from higher- to lower-potential groups, so the final stage still uses **all data**.
> 2. **Online-Unmerge:** Similar to Online-Merge, but each stage trains only on a **single unmerged group**, moving from high- to low-potential groups. This setup represents a typical online curriculum learning scenario.
>
> | Method | Step 0 | 50 | 100 | 150 | 200 | 250 | 300 | 350 | 400 | 450 | 500 | 550 | 600 |
> |---------|--------|----|------|------|------|------|------|------|------|------|------|------|------|
> | Offline | 60.6 | 61.3 | 62.5 | 65.5 | 67.5 | 68.5 | 69.4 | 71.1 | 71.7 | 71.9 | 72.1 | 72.4 | 72.3 |
> | Online-Merge | 60.6 | 61.3 | 62.5 | 65.5 | 67.7 | 68.9 | 69.7 | 71.9 | 72.3 | 72.4 | 72.3 | 72.4 | 72.5 |
> | Online-Unmerge | 60.6 | 61.3 | 62.5 | 65.5 | 66.9 | 67.9 | 68.5 | 70.0 | 70.9 | 71.1 | 71.2 | 71.3 | 71.3 |
>
> These three methods are different after the training of stage 1 (step 150). Our results show that **offline and online-merge training achieve similar performance** in later stages. Although the online approach converges faster, it requires **higher inference costs** due to repeated repartitioning. The online-unmerge variant performs worse overall, indicating that training on the entire dataset (as in the offline or merged online setting) better captures overall data utility. In summary, the **offline training strategy** achieves **comparable effectiveness** to online methods while being **more efficient** computationally.
>
> ---
>
> ### **Q6. Bernoulli reward assumption**
>
> **Reviewer’s concern:**
> Why assume a Bernoulli reward distribution? Given the diversity of tasks and the complexity of model parameters, the reward distribution of samples may be complex and difficult to represent explicitly.
>
> **Response:**
> In our formulation, the Bernoulli assumption **does not model reward distributions across tasks or questions**, but rather the reward distribution of **multiple rollouts for the same question under identical LLM sampling conditions** (e.g., temperature, model type). These rollouts are used to compute the advantage within our GRPO framework.
> Each rollout (i.e., sampled response) receives a binary terminal reward: 1 if the answer is correct and 0 otherwise. Since these rollouts are independent with a correctness probability p, the reward variable **naturally follows a Bernoulli distribution**.
> This assumption allows the theoretical derivations in Equations 4.3–4.5 to be analytically tractable and explains two key insights:
> - Policy gradients vanish when most samples are either too easy or too hard.
> - The improvement potential is maximized when p=0.5.
>
> ---
>
> ### **Q7. Criteria for adding lower-potential groups**
>
> **Reviewer’s concern:**
> What thresholds define the inclusion of new lower-potential groups? How large is each stage’s dataset?
>
> **Response:**
> We apologize for omitting these details earlier. As clarified in the revised paper (**Page 6, Line 316**), the dataset is divided into **four equal-sized groups** based on improvement potential (IP) values:
>
> | Group | IP Range |
> |--------|-----------|
> | 1 | [0.0, 0.32] |
> | 2 | [0.32, 0.48] |
> | 3 | [0.48, 0.64] |
> | 4 | [0.64, 1.0] |
>
> Each stage progressively adds one group (from high to low potential), ensuring balanced data volumes and smooth curriculum transitions.
>
> ---
>
> **We sincerely thank the reviewer once again for your valuable feedback and are happy to discuss any remaining questions.**

---

> > ### Comment · Reviewer_Rp2m · 2025-11-25
> > **Official Comment from Reviewer Rp2m**
> >
> > Thank you for the author's response. I also read the author's revised version, which appear convincing. I believe that including other algorithms beyond GRPO would further strengthen the paper's arguments and broaden its scope. That said, I will raise my score to 6 "weak accept". I hope the authors can incorporate these conclusions into the revised version.

---

> ### Author Response · Authors · 2025-11-27
> **Thank you for raising the score and appreciation into our work**
>
> Thank you for raising the score and the helpful suggestions. Your suggested experiments truly improve the quality of the paper and make the insights more solid. We will definitely incorporate these conclusions into the revised version.
>
> Best wishes,
> Authors of R1-Code-Interpreter

---

### Author Response · Authors · 2025-11-29
**Summary of Review and Rebuttal Status for R1-Code-Interpreter**

Dear Area Chair,

Due to the unexpected leakage of author and reviewer information, we understand that the author–reviewer discussion period has been terminated, all score changes have been reverted, and your recommendation will be based on the **initial reviews and the existing discussion**. To assist your assessment, we summarize the review status below.

### Reviewer Rp2m
All concerns from Reviewer Rp2m have been fully addressed and the overall score has been raised to 6, as shown in the discussion contents. Although the ICLR committee has reverted all score changes, Reviewer Rp2m **raised the score from 4 to 6 (“weak accept”)** prior to the leakage incident on *25 Nov 2025*, well before the event on *27 Nov 2025*.
In the initial review (08 Nov 2025), the reviewer explicitly stated willingness to raise the score if concerns were addressed. Therefore, we believe the **score increase is reliable and reflective of the reviewer’s updated evaluation**.

### Reviewer Eyn2
Reviewer Eyn2 maintained the score of **8 (“accept”)**, stating that **all concerns were resolved** following our rebuttal.

### Reviewer HYQj
Reviewer HYQj had only one remaining question, which we believe stems from a misunderstanding of the experimental setting. The reviewer appears to be asking for comparisons between our multi-turn Code Interpreter setting and the All-Text / All-Code settings under equal data and base-model conditions.
As explained in our final discussion, **these experiments are already included in the paper**, and the reviewer acknowledged our main contribution of **training a general Code Interpreter across multiple tasks and domains**. Although the reviewer did not have the chance to respond further due to the early termination of discussion, we firmly believe the reviewer would **raise the score** based on the clarified evidence.

---

**In summary, all three reviewers expressed agreement with the acceptance of this paper**, based on the addressed concerns, updated discussions, and stated intentions.

We kindly hope that you could consider these contexts when making your final recommendation.

Best regards,
Authors of R1-Code-Interpreter

---

### Meta-Review · Area_Chair_QXyn · 2025-12-06

**Summary:**

This paper trains LLM to use hybrid code and text reasoning to solve reasoning tasks of different domains. The training starts with an SFT stage followed by RL. The authors proposed a curriculum training on RL to overcome the learning stagnation due to diverse reasoning domains. The curriculum works on training first on samples with higher improvement potential, and gradually including more samples until the full training dataset is covered. The method shows improvements when applied on several open source models as compared to several baselines.

Reviewer major concerns are if the proposed approach work across different RL methods and fairness of evaluation. Both are addressed with authors adding additional experiments.

**Reviewer Concerns:**

The evaluation faireness as compared to baseline seems to be a misunderstanding from reviewers. The authors has presented results that supports the claim

**Reviewer Scores:**

Rp2m 4 -> 6, asked additional methods on RL is added, self mentioned raise to 6
Eyn2 8 -> 8, no particular concern, stated will keep the score
HYQj 4 -> 4+, concerns mostly addressed, major concern seems to be a misunderstanding

---

### Decision · Program_Chairs · 2026-01-26

Accept (Poster)